# CAMO: Convergence-Aware Multi-Fidelity Bayesian Optimization

## Abstract

Multi-fidelity Bayesian Optimization (MFBO) has emerged as a powerful approach for optimizing expensive black-box functions by leveraging evaluations at different fidelity levels. However, existing MFBO methods often overlook the convergence behavior of the objective function as fidelity increases, leading to inefficient exploration and suboptimal performance. We propose CAMO, a novel Convergence-Aware Multi-fidelity Optimization framework based on Fidelity Differential Equations (FiDEs). CAMO explicitly captures the convergence behavior of the objective function, enabling more efficient optimization. We introduce two tractable forms of CAMO: an integral Automatic Relevance Determination (ARD) kernel and a data-driven Deep Kernel. Theoretical analysis demonstrates that CAMO with the integral ARD kernel achieves a tighter regret bound compared to state-of-the-art methods. Our empirical evaluation on synthetic benchmarks and real-world engineering design problems shows that CAMO consistently outperforms existing MFBO algorithms in optimization efficiency and solution quality, with up to 4x improvement in optimal solution. This work establishes a foundation for tractable convergence-aware MFBO and opens up new avenues for research in this area.

## 1 Introduction

Bayesian Optimization (BO) has emerged as a powerful approach for optimizing expensive black-box functions, finding applications in various domains such as engineering design and scientific experimentation (Brochu et al., 2010). BO leverages a surrogate model, typically a Gaussian Process (GP), to approximate the objective function and guide the search towards promising regions using an acquisition function. This approach is particularly effective when function evaluations are costly or time-consuming.

In practical applications, it is often possible to obtain evaluations of the objective function at varying levels of fidelity or accuracy, with lower fidelity evaluations typically being less costly. For example, in engineering design, simplified simulations with coarser meshes or fewer iterations can provide approximate results at a fraction of the cost of high-fidelity simulations. This multi-fidelity nature of many real-world problems has led to the development of Multi-Fidelity Bayesian Optimization (MFBO) methods, which aim to leverage information from lower-fidelity evaluations to reduce the overall optimization cost and improve convergence speed (Kandasamy et al., 2017; Wu et al., 2020).

While MFBO has shown promise, existing methods often model the objective function as a black-box across fidelity levels, ignoring the inherent convergence behavior as fidelity increases. This limitation can lead to inefficient exploration of the search space, particularly in costly high-fidelity regions, resulting in suboptimal performance and a lack of theoretical guarantees when the fidelity can approach infinity (a common scenario in numerical simulations where the number of elements can be arbitrarily large or the time step arbitrarily small).

To address these challenges, we propose CAMO (Convergence-Aware Multi-fidelity Optimization), a novel MFBO framework based on Fidelity Differential Equations (FiDEs). CAMO explicitly captures the convergence behavior of the objective function through rigorous theoretical analysis, enabling more efficient optimization across fidelity levels. Our approach allows for automatic adaptation to varying fidelity levels without extensive manual intervention, making it particularly suitable for practical MFBO scenarios where the highest and lowest fidelity levels may be unknown or difficult to specify. The main contributions of this work are as follows:

1. We introduce a convergence-aware multi-fidelity surrogate model based on FiDEs, with rigorous convergence conditions to capture the behavior of the objective function across fidelity levels.

2. We propose two practical implementations of our model: a nonstationary integral Automatic Relevance Determination (ARD) kernel and a flexible deep kernel.

3. We provide theoretical analysis demonstrating that CAMO with the integral ARD kernel achieves a tighter regret bound compared to the state-of-the-art BOCA method, offering theoretical guarantees of superior performance.

4. We conduct a comprehensive empirical evaluation of CAMO against state-of-the-art MFBO methods on both synthetic benchmarks and real-world engineering problems, demonstrating consistent superior performance in terms of optimization efficiency and solution quality.

5. We provide insights into the effectiveness of various multi-fidelity surrogate models and acquisition functions, contributing to a better understanding of MFBO strategies.

By establishing a foundation for convergence-aware MFBO, this work opens up new avenues for research in multi-fidelity optimization and provides a powerful tool for practitioners dealing with expensive multi-fidelity optimization problems.

## 2 RELATED WORK

Bayesian Optimization (BO) is a powerful framework for optimizing expensive black-box functions (Shahriari et al., 2015). It leverages a surrogate model to approximate the objective function and an acquisition function to guide the search towards promising regions. Gaussian Processes (GPs) are commonly used as surrogate models due to their ability to quantify uncertainty and provide smooth interpolations (Rasmussen and Williams, 2006). Popular acquisition functions include Expected Improvement (EI) (Mockus, 1978), Upper Confidence Bound (UCB) (Srinivas et al., 2009), and Probability of Improvement (PI) (Kushner, 1964). These acquisition functions balance exploration and exploitation to efficiently search for the global optimum.

Multi-Fidelity Bayesian Optimization (MFBO) extends standard BO to leverage evaluations at different fidelity levels, aiming to reduce the overall optimization cost and improve convergence speed. Early works in MFBO focused on problems with discrete fidelity levels, where the objective function can be evaluated at a finite number of fidelity levels. Huang et al. (2006) proposed Sequential Kriging Optimization (SKO), which employs a hierarchical kriging/GP to capture the correlations between different fidelity levels. Kandasamy et al. (2016a) proposed the popular Multi-Fidelity Gaussian Process Upper Confidence Bound (MF-GP-UCB), which employs a simple GP as the surrogate and a modified UCB acquisition function to balance exploration and exploitation in the multi-fidelity setting. To improve model capacity for complex multi-fidelity problems, several works have focused on developing efficient MF surrogates for problems with discrete fidelity levels. Le Gratiet and Garnier (2014) proposed Recursive Co-Kriging, which builds a hierarchy of GP models, each modeling the residual between successive fidelity levels. Perdikaris et al. introduced Nonlinear Autoregression (NAR), which employs several GPs to capture complex nonlinear relationships between fidelity levels. Cutajar et al. (2019) proposed the Deep Multi-Fidelity Gaussian Process, which employs a deep GP to learn nonlinear mappings between fidelity levels.

In many real-world applications, the fidelity level is a continuous variable (e.g., the mesh size or the tolerance level in numerical simulations). This has led to the development of continuous fidelity MFBO methods. Kandasamy et al. (2017) proposed Bayesian Optimization with Continuous Approximations (BOCA), which models the objective function as a GP with a continuous fidelity variable and employs a two-stage acquisition function to balance the exploration-exploitation trade-off. Poloczek et al. (2017) introduced the Multi-Information Source Optimization (MISO) framework, which models the objective function as a linear combination of GPs, each corresponding to a different information source (fidelity level). Klein et al. (2017) proposed FABOLAS, which employs a modified GP with a linear fidelity kernel and MF Entropy Search as the acquisition function and strategy in the multi-fidelity setting. Wu and Frazier (2018) proposed a continuous fidelity knowledge gradient (cfKG), which extends the knowledge gradient acquisition function to the continuous fidelity setting with a simple GP surrogate. Li et al. (2020) proposed Deep Neural Network Multi-Fidelity Bayesian Optimization (DNN-MFBO) with a fidelity-wise Gauss-Hermite quadrature and moment-matching mutual information acquisition function. These works have showcased the benefits of continuous fidelity MFBO over the discrete counterparts. However, most practical GP-based methods lack

insights into the convergence behavior of the objective function across fidelity levels, leading to the over-exploration of costly regions and suboptimal performance. While a DNN might be able to capture the complex relationships between fidelity levels, it often faces challenges of computational complexity, choice of network architectures, and overfitting. DNN does not seem to be a stable model for BO where data is extremely limited.

Recent advancements in multi-fidelity modeling propose a new concept of fidelity differential equations (FiDE) (Xing et al., 2024), which is then modeled by NeuralODE (Li et al., 2022) or a continuous autoregression (CAR) (Xing et al., 2024) to capture the convergence behavior of the objective function. Despite the powerful model capacity of NeuralODE, it is challenging to train (up to 10,000x training time compared to CAR). Thus, our work will build upon the FiDE concept and propose a tractable and scalable MF surrogate.

## 3 BACKGROUND

### 3.1 PROBLEM FORMULATION

We consider the problem of optimizing a continuous function $f : \mathcal{X} \times \mathcal{T} \to \mathbb{R}$, where $\mathcal{X} \subseteq \mathbb{R}^d$ is a compact set representing the domain of the optimization variables and $\mathcal{T} = [t_0, T) \subset \mathbb{R}$ is the range of fidelity levels, with $t_0$ being the lowest fidelity and $T$ being the highest fidelity, which can be infinite. The function $f(x, t)$ represents the performance of a system or design characterized by variables $x$ at fidelity level $t$.

In the context of MFBO, we assume that evaluating $f(x, t)$ at a higher fidelity level $t$ yields more accurate results but is also more costly in terms of computational resources or time, denoted as $c(t)$, where $c(t)$ is assumed a monotonically increasing function. The goal is to find the optimal solution $x^* \in \mathcal{X}$ that maximizes the objective function at the highest fidelity level $T$, i.e., $\arg\max f(x, T)$, while minimizing the total cost of evaluations.

### 3.2 MULTI-FIDELITY SURROGATE

MFBO requires a surrogate to model $f(\mathbf{x}, t)$, which is often assumed to be a zero mean GP (i.e., $f(\mathbf{x}, t) \sim \mathcal{GP}\left(0, k(\mathbf{x}, t, \mathbf{x}', t')\right)$) with a separable kernel (i.e., $k(\mathbf{x}, t, \mathbf{x}', t') = k_x(\mathbf{x}, \mathbf{x}') \cdot k_t(t, t')$).

To capture the complex relationships between fidelities and design variables, Li et al. (2022) propose a general Fidelity Differential Equation (FiDE) as the fidelity surrogate model,

$$\dot{y}(\mathbf{x}, t) = \phi(\mathbf{x}, t, y(\mathbf{x}, t)), \tag{1}$$

where $\dot{y}(\mathbf{x}, t)$ is derivative w.r.t fidelity $t$, and $\phi(\mathbf{x}, t, y)$ is a general function describing the system dynamics with respect to fidelity. This formulation leads to a NeuralODE (Chen et al., 2018) solution, which allows modeling complex systems leveraging the power of deep learning.

## 4 METHODOLOGY

To conduct MFBO efficiently without over-exploration of costly yet low-return regions, it is crucial to ensure that the surrogate $y(\mathbf{x}, t)$ converges as the fidelity variable approach $T$. This is particularly important when the highest fidelity level is infinite. For instance, it is crucial in finite element analysis to find a proper mesh size that can accurately capture the behavior of the system within a feasible computational cost.

### 4.1 CONVERGENCE CONDITIONS FOR FIDE

To ensure the convergence of $y(\mathbf{x}, t)$, We start by exploring the convergence conditions for FiDE:

**Lemma 1.** *Convergence Conditions for FiDE:* $\lim_{t \to T} y(\mathbf{x}, t) = y(\mathbf{x}, T)$ *can be guaranteed if there exists a continuously differentiable Lyapunov function $V : \mathcal{X} \times \mathbb{R} \times [0, T) \to \mathbb{R}$ such that:*

1. $V(\mathbf{x}, y, t) > 0, \forall (\mathbf{x}, y, t) \in \mathcal{X} \times \mathbb{R} \times [0, T)$ *with* $y \neq y(\mathbf{x}, T)$*, and* $V(\mathbf{x}, y(\mathbf{x}, T), t) = 0$,

2. $\frac{\partial V(\mathbf{x}, y, t)}{\partial y} \phi(\mathbf{x}, t, y) + \frac{\partial V(\mathbf{x}, y, t)}{\partial t} < 0, \forall (\mathbf{x}, y, t) \in \mathcal{X} \times \mathbb{R} \times [0, T)$ *with* $y \neq y(\mathbf{x}, T)$.

See Appendix B for the proof. This lemma provides a general framework for deriving convergence conditions using a Lyapunov function. The positive-definite property of the Lyapunov function ensures that it represents a measure of the distance from the steady-state solution, while the negative-definite derivative condition guarantees that the function decreases along the system trajectories, leading to convergence.

To apply this lemma, one needs to find a suitable Lyapunov function $V(\mathbf{x}, y, t)$ that satisfies the given conditions for the specific form of $\phi(\mathbf{x}, t, y)$. The choice of the Lyapunov function depends on the problem structure and the desired convergence properties. For example, a common choice of Lyapunov function is the quadratic form, $V(\mathbf{x}, y, t) = (y(\mathbf{x}, t) - y(\mathbf{x}, T))^2$, which represents the squared distance from the steady-state solution $y(\mathbf{x}, T)$. If the derivative condition holds for this Lyapunov function, i.e., if

$$2\left(y(\mathbf{x}, t) - y(\mathbf{x}, T)\right)\phi(\mathbf{x}, t, y) + \frac{\partial V(\mathbf{x}, y, t)}{\partial t} < 0, \tag{2}$$

for all $(\mathbf{x}, y) \neq (\mathbf{x}, y(\mathbf{x}, T))$ and $t \geq 0$, then the solution $y(\mathbf{x}, t)$ converges to $y(\mathbf{x}, T)$ as $t \to T$, for each $\mathbf{x} \in \mathcal{X}$. The challenge is that real $\phi(\mathbf{x}, t, y)$ is unknown and difficult to model. One possible solution is to impose a regularization loss term that enforces the convergence conditions, similar to the physics-informed neural networks (PINNs) (Raissi et al., 2019),

$$\mathcal{L}_{\text{Lyapunov}} = \max\left(0, \frac{\partial V}{\partial y}\phi\theta(\mathbf{x}, t, y) + \frac{\partial V}{\partial t}\right). \tag{3}$$

However, even if we can solve this problem, the complexity is likely to overwhelm the complexity of the original MF optimization problem. A more tractable formulation is needed.

### 4.2 CONVERGENCE CONDITIONS FOR LINEAR FIDE

To address the over-general formulation of Eq. (1), Xing et al. (2024) to consider a more tractable case by assuming that the function $\phi(\mathbf{x}, t, y(\mathbf{x}, t))$ admits a linear form, leading to the linear FiDE:

$$\dot{y}(\mathbf{x}, t) = \beta(\mathbf{x}, t)y(\mathbf{x}, t) + u(\mathbf{x}, t). \tag{4}$$

However, the original work does not guarantee convergence for practical applications, which we will resolve in this work by exploring its convergence conditions:

**Lemma 2.** *Convergence conditions for Linear FiDE:* $\lim_{t \to T} y(\mathbf{x}, t) = -\frac{u^*(\mathbf{x})}{\beta(\mathbf{x}, T)}, \forall \mathbf{x} \in \mathcal{X}$, *can be guaranteed if: (1)* $\beta(\mathbf{x}, t) < 0, \forall(\mathbf{x}, t) \in \mathcal{X} \times [0, T)$, *and (2)* $\lim_{t \to T} u(\mathbf{x}, t) = u^*(\mathbf{x}), \forall \mathbf{x} \in \mathcal{X}$,

See Appendix C for a detailed proof. The lemma states the key convergence conditions and the steady-state solution for the Linear FiDE. The first condition ensures that the system dynamics, governed by $\beta(\mathbf{x}, t)$, are stable, while the second condition requires the input function $u(\mathbf{x}, t)$ to converge to a desired target value $u^*(\mathbf{x})$. For example, a simple choice of $\beta(\mathbf{x}, t)$ could be a constant negative value, i.e., $\beta(\mathbf{x}, t) = -\beta$, where $\lambda > 0$. This choice leads to an exponential convergence of the solution to the steady state, with a convergence rate determined by $\lambda$. A larger value of $\lambda$ results in faster convergence, while a smaller value leads to slower convergence.

Alternatively, $\beta(\mathbf{x}, t)$ could be designed as a function of the input $\mathbf{x}$ and fidelity $t$ to adapt the convergence behavior based on the problem-specific requirements. For instance, $\beta(\mathbf{x}, t)$ could be modeled as a neural network, allowing for more flexible and data-driven convergence dynamics.

### 4.3 TRACTABLE AUTOREGRESSIVE GAUSSIAN PROCESS MULTI-FIDELITY SURROGATE

Building upon the linear FiDE, we now introduce a tractable multi-fidelity model that captures the convergence behavior while maintaining the probabilistic framework of Gaussian Processes for the BO task. We propose two variants of this model: an Integral Automatic Relevance Determination (ARD) kernel and a Deep Kernel. First, let us consider the Integral ARD kernel. Given GP priors:

$$y(\mathbf{x}, t_0) \sim \mathcal{GP}(0, k^0(\mathbf{x}, \mathbf{x}')), \quad u(\mathbf{x}, t) \sim \mathcal{GP}(0, k^u(\mathbf{x}, t, \mathbf{x}', t')), \tag{5}$$

we can derive (see Appendix D.1) that the joint mean is zero and covariance functions $k(\mathbf{x}, t, \mathbf{x}', t') =$

$$e^{\int_{t_0}^{t} \beta(\mathbf{x}, s)ds}e^{\int_{t_0}^{t'} \beta(\mathbf{x}', s)ds}k^0(\mathbf{x}, \mathbf{x}') + \int_{t_0}^{t}\int_{t_0}^{t'} e^{\int_{s}^{t} \beta(\mathbf{x}, u)du}e^{\int_{s'}^{t'} \beta(\mathbf{x}', u)du}k^u(\mathbf{x}, s, \mathbf{x}', s')dsds'. \tag{6}$$

Working with the integral in Eq. (6) seems challenging. At least numerical integration is required, i.e., $\frac{1}{M}\sum_{i,j}^{M} e^{-\beta(t_i + t_j - 2t_0)}k^u(\mathbf{x}, \tau_i, \mathbf{x}', \tau_j)$, where $t_i$ and $t_j$ are $M$ are random samples from $[t_0, t]$ and $[t_0, t']$, respectively.

Surprisingly, the integral in Eq. (6) can be analytically evaluated under certain conditions. We can derive closed-form solutions for various kernel choices and forms of $\beta(\mathbf{x}, t)$. In particular, either 1) constant $\beta(\mathbf{x}, t) = \beta$ with different stationary kernels (i.e., Matérn, periodic, random Fourier features) for $k^u(\mathbf{x}, s, \mathbf{x}', s')$ or 2) polynomial $\beta(\mathbf{x}, t)$ with a separable kernel $k^u(\mathbf{x}, t, \mathbf{x}', t')$ can lead

to a closed-form solution for the covariance function. The reader is referred to the Appendix D for the resulting kernel functions of $k(\mathbf{x}, t, \mathbf{x}', t')$ and detailed proofs for various kernel choices.

**Integral ARD**: In this work, we focus on the simple case to highlight that superior performance stems from convergence-awareness rather than complex kernel design. We consider a constant $\beta$ and an ARD kernel for $k^u(\mathbf{x}, t, \mathbf{x}', t')$, which leads to our Integral ARD kernel:

$$k(\mathbf{x}, t, \mathbf{x}', t') = e^{\beta(t-t_0)}e^{\beta(t'-t_0)}k^0(\mathbf{x}, \mathbf{x}') + k_{\mathbf{x}}^u(\mathbf{x}, \mathbf{x}')I(t, t'), \tag{7}$$

where $k^u$ is the ARD kernel function, $I(t, t')$ is the integral over the ARD kernel and can be analytically evaluated using the properties of the Gaussian integral. The detailed derivation of $I(t, t')$ was first given by Álvarez et al. (2020) for GP of a linear dynamical system where $t$ becomes time. The explicit form of $I(t, t')$ is:

$$I(t, t') = \sqrt{\frac{\pi}{2}}\ell e^{\beta(t+t'-2t_0)}\left[\text{erf}\left(\frac{t-t_0}{\sqrt{2}\ell}\right) + \text{erf}\left(\frac{t'-t_0}{\sqrt{2}\ell}\right)\right]$$
$$- \ell^2 e^{\beta(t+t'-2t_0)+\frac{\beta^2\ell^2}{2}}\left[e^{-\beta(t-t_0)}\text{erf}\left(\frac{t-t_0-\beta\ell^2}{\sqrt{2}\ell}\right) + e^{-\beta(t'-t_0)}\text{erf}\left(\frac{t'-t_0-\beta\ell^2}{\sqrt{2}\ell}\right)\right], \tag{8}$$

where $\text{erf}(\cdot)$ denotes the error function. This kernel is non-stationary with a length scale increasing as $t$ increases, suggesting a stronger correlation when the fidelity level is closer to the highest level, which is consistent with our intuition and observations in practice. An intuitive illustration is also given in Figure 1. Moreover, the lowest-fidelity $t_0$ can be learned from the data, enabling automatic setting of the lowest fidelity, below which any execution is not meaningful for MFBO.

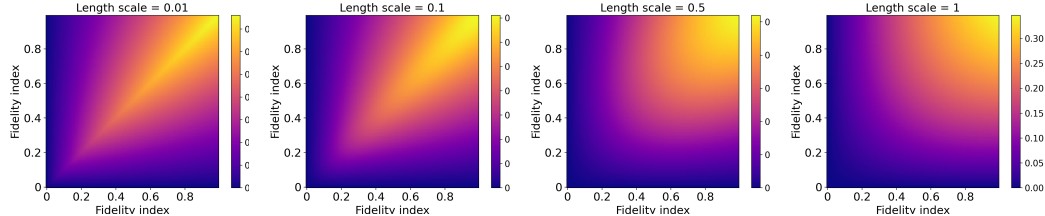

Figure 1: Effect of the length scale parameter $\ell$ on integral ARD kernel: high fidelity are highly correlated whereas the low-fidelity correlations are significantly influenced by $\ell$.

**Deep Kernel**: Inspired by the philosophy of deep learning and its success, we can simply impose a deep kernel $k_{\text{DKL}}(s, s')$ to approximate the integral term in Eq. (6). This approach allows us to learn a flexible and data-driven representation of the integration term, capturing the complex relationships between the input features and the convergence behavior of the system. As a price, we lose explicit control of the convergence.

Whichever Integral ARD or Deep Kernel we use, the additive kernel structure (7) allows us to train the MF surrogate with complexity $\mathcal{O}(\sum_{f=1}^F N_f^3)$ not $\mathcal{O}\left((\sum_{f=1}^F N_f)^3\right)$, where $N_f$ is the number of data points at fidelity $f$. For more details, see Appendix F.

### 4.4 OPTIMIZATION STRATEGIES FOR CONTINUOUS FIDELITY ACQUISITION FUNCTIONS

In MFBO, the main difference compared to standard BO lies in the optimization strategy used to select the candidates and the fidelity level to evaluate. The optimization strategies need to account for the fidelity levels and the trade-off between the cost and accuracy of evaluations. MF-GP-UCB (Kandasamy et al., 2016a) extend UCB as

$$\alpha_{\text{MF-UCB}}(x, t) = (\mu(x, t) + \beta\sigma(x, t))/\sqrt{c(t)}, \tag{9}$$

which is maximized to generate the candidate design $\mathbf{x}$ and execution fidelity level $t$. BOCA (Kandasamy et al., 2017) use the same UCB acquisition but with a cost-weighted optimization strategy and optimize $\mathbf{x}$ first, conditioned on which the optimal execution $t$ is found.

cfKG (Wu et al., 2020) is a continuous extension of the discrete-fidelity KG. The optimization strategy in cfKG is to select the next point and fidelity level that maximizes the cost-weighted KG acquisition function:

$$\alpha_{\text{cfKG}}(x, t) = \mathbb{E}[\max_x \mu^+(x, T) - \max_x \mu(x, T)]/c(t). \tag{10}$$

Other SOTA MF acquisition includes MF Expected Improvement (EI) used by FABOLAS (Klein et al., 2017) and approximated mutual-information (MI) by MF-DNN (Li et al., 2020). Most acquisitions are independent of the MF model and can be used in any MFBO. We tested CAMO (and other surrogates) with different acquisition functions and found that BOCA acquisition is the most effective for CAMO and most other surrogates despite its simplicity. In practice, the choice of the optimization strategy also depends on the specific problem characteristics and the available computational resources.

## 5 THEORETICAL ANALYSIS

We provide theoretical analysis of the proposed CAMO, focusing on the regret bounds for BOCA using CAMO with Integral ARD kernel.

**Theorem 5.1** (**Regret of BOCA**). *Let $f_*$ be the optimal value of the objective function $f(\mathbf{x}, T)$, and let $S(\Lambda)$ be the simple regret of the BOCA algorithm with a batch of evaluations $\Lambda$. Then, with probability at least $1 - \delta$, the simple regret of BOCA satisfies:*

$$S(\Lambda) \lesssim \sqrt{\frac{\Psi_{n_\Lambda}(\mathcal{X}\rho)}{n\Lambda}} + \sqrt{\frac{\Psi_{n_\Lambda}(\mathcal{X})}{n_\Lambda^{1-\alpha}}}, \tag{11}$$

*where $\Psi_{n_\Lambda}(\mathcal{X}\rho)$ and $\Psi n_\Lambda(\mathcal{X})$ are the maximum information gains for the surrogate kernel over the sets $\mathcal{X}_\rho$ and $\mathcal{X}$, respectively, and $\alpha \in (0, 1)$ is a constant that depends on the kernel.*

Most queries to $f(\mathbf{x}, T)$ will be confined to a small subset of the domain $\mathcal{X}$ which contains the optimum $\mathbf{x}_*$. More precisely, after capital $\Lambda$, for any $\alpha \in (0, 1)$, Kandasamy et al. (2017) show that there exists $\rho > 0$ such that the number of queries outside the following set $\mathcal{X}\rho$ is less than $n\Lambda^\alpha$:

$$\mathcal{X}\rho = \left\{ \mathbf{x} \in \mathcal{X} : f - f(\mathbf{x}, T) \leq 2\rho\sqrt{\kappa_0 \xi(\sqrt{p})} \right\}, \tag{12}$$

where $p$ is the effective distance from the maximum fidelity $T$, $\kappa_0$ is a constant, and $\xi(\cdot)$ is a function that depends on the kernel. Assuming the difference between fidelities $k_t(t, t')$ forms a SE kernel, we have

$$\mathcal{X}\rho \approx \left\{ \mathbf{x} \in \mathcal{X} : f * - f(\mathbf{x}, T) \leq 2\rho\sqrt{p/\ell} \right\}, \tag{13}$$

where $\ell$ is the length scale of the SE kernel. When $\ell$ is large and $f$ is smooth across the fidelity space $\mathcal{T}$, $\mathcal{X}\rho$ is small as the right side of the inequality is smaller. As BOCA confines most of its evaluations to this small set containing $\mathbf{x}*$, it can achieve much better regret than standard GP-UCB.

In the regret bound (11), the latter term vanishes fast due to the $n_\Lambda^{(1-\alpha)/2}$ dependence. Comparing this with the regret bound of GP-UCB, we see that BOCA outperforms GP-UCB by a factor of $\sqrt{\Psi_{n_\Lambda}(\mathcal{X}\rho)/\Psi n_\Lambda(\mathcal{X})} \lesssim \sqrt{\text{vol}(\mathcal{X}\rho)/\text{vol}(\mathcal{X})}$ asymptotically. If $f$ is smooth across the fidelity space, $\mathcal{X}\rho$ is small, and the gains over GP-UCB are significant. If $f$ becomes less smooth across $\mathcal{T}$, the bound decays gracefully, but BOCA is never worse than GP-UCB up to constant factors.

Our work derives a more natural kernel in Eq. (8) with convergence-awareness for the multi-fidelity surrogate. With some simplifications (See Appendix E), the integral ARD kernel leads to

$$\mathcal{X}_\rho \approx \left\{ x \in \mathcal{X} : f_* - f(x) \leq 2\rho\sqrt{\kappa_0} \exp\left(-\frac{p^{1+\alpha/2}}{2\gamma^{1/2}}\right) \right\}, \tag{14}$$

where $p$ is the effective distance from the maximum fidelity, and $\kappa_0$, $\gamma$, and $\alpha$ are constants that depend on the kernel parameters. The faster decay of the approximation term in $\mathcal{X}\rho$ leads to a smaller $\Psi n_\Lambda(\mathcal{X}_\rho)$ compared to the SE kernel case, resulting in a tighter first term in the regret bound. Additionally, the second term in the regret bound (11) has a smaller exponent $1 - \alpha/2$. Thus, the regret bound is tighter than BOCA.

In hindsight, the use of the specific non-stationary integral kernel can lead to a tighter regret bound than using the SE kernel because it allows $\ell$ to increase as we approach the maximum fidelity, whereas the original SE kernel must have a fixed $\ell$ across the fidelity space. With the convergence-awareness, CAMO enables efficient exploration of the fidelity space, leading to faster convergence to the optimum.

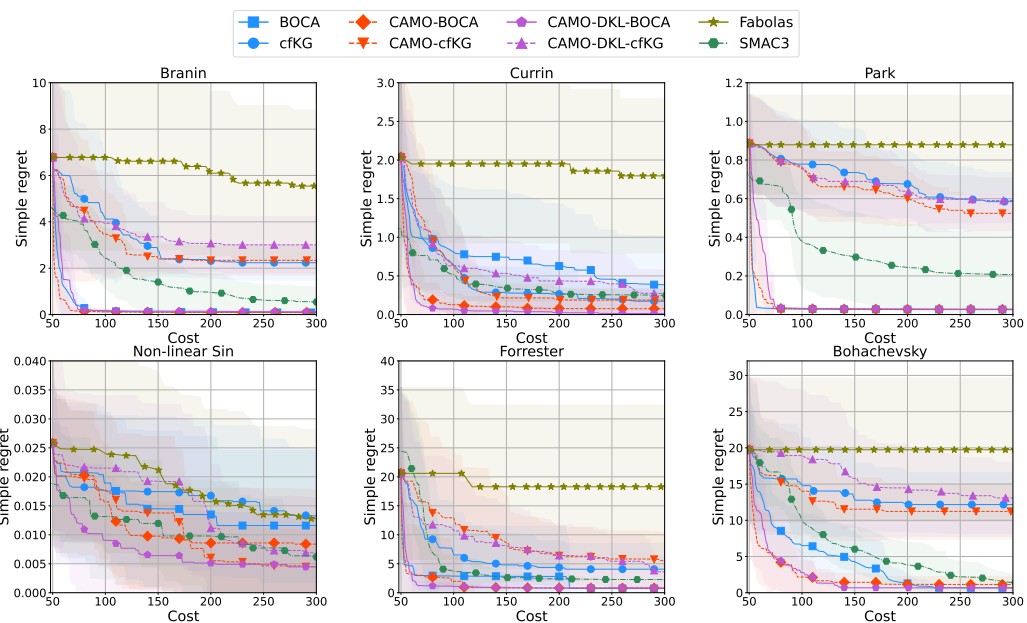

Figure 2: Continuous MFBO for Branin, Currin, Park, Non-linear-sin, Forester, and Bohachevsky.

# 6  EXPERIMENT

We assess CAMO on synthetic benchmarks, including continuous and discrete MFBO tasks, and real-world engineering design tasks.

**Competing Methods.** We compare CAMO with the SOTA continuous MFBO methods including (1) BOCA Kandasamy et al. (2017), (2) Fabolas Klein et al. (2017), and (3) SMAC3 Domhan et al. (2015). To showcase the superiority of CAMO on discrete fidelity tasks, we also assess CAMO against the SOTA discrete MFBO methods based on combinations of the SOTA discrete surrogates, namely, (1) AR (Kennedy, 2000), (2) ResGP (Xing et al., 2021), and (3) GP (Rasmussen and Williams, 2006), and the SOTA discrete acquisition functions, namely, (1) MF-UCB (Kandasamy et al., 2016b), (2) MF-EI (Huang et al., 2006), and (3) cfKG (Wu and Frazier, 2018).

**Setting and Results.** We assess CAMO with the integral ARD kernel as in Eq. (7) and a Deep kernel (DKL), which always use a 4-layer MLP with $d \times 4$ hidden units and leaky ReLU activation functions. Except for DNN-MFBO [1], Fabolas [2], and SMAC3 [3] which use their original implementations and default settings, all other methods are implemented by us using Pytorch. All GPs use the ARD Kernel for a fair comparison if not stated otherwise. Each model is updated for 200 steps using Adam optimizer with a learning rate of $0.01$, which is sufficient for model convergence. For each task, we randomly select 10 low-fidelity designs and 4 high-fidelity designs to form the initial training set. Each algorithm is run until convergence. We repeat the experiments 20 times with random seeds and report the mean and standard deviation (thus the markers do not indicate the actual evaluations). Figures showing the actual optimization progress are preserved in Appendix I. The optimization performance is measured by simple regret (SR), defined as the difference between global optimum and the best-queried design so far: $SR_i = \max f(\mathbf{x}, T) - \max_{j<i} f(\mathbf{x}_j, T)$. All experiments are run on a workstation with AMD 7800x CPU, Nvidia RTX4080 GPU, and 32GB RAM.

## 6.1  SYNTHETIC BENCHMARK EVALUATION

We first evaluated CAMO on three canonical continuous fidelity tasks (Xiong et al., 2013): Park function, Currin function, and Branin, and three synthesized continuous fidelity tasks from tow-fidelity benchmarks (Ankenman et al., 2008): nonlinear sin, Forrester, and Bohachevsky function by defining the continuous fidelity as $f(\mathbf{x}, t) = (1 - w(t))f_{low}(\mathbf{x}) + w(t)f_{high}(\mathbf{x})$ with $w(t) = \ln(9t + 1)$. The details of the benchmarks are in Appendix G. The query costs are set to $c(t) = 10^t$.

---

[1] https://github.com/shib0li/DNN-MFBO [2] https://github.com/rickie95/fabolas [3] https://github.com/automl/SMAC3

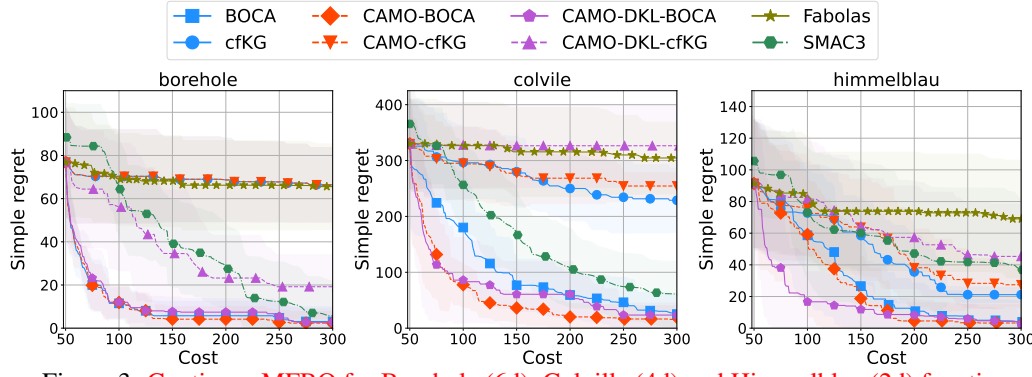

Figure 3: Continues MFBO for Borehole (6d), Colville (4d) and Himmelblau (2d) function

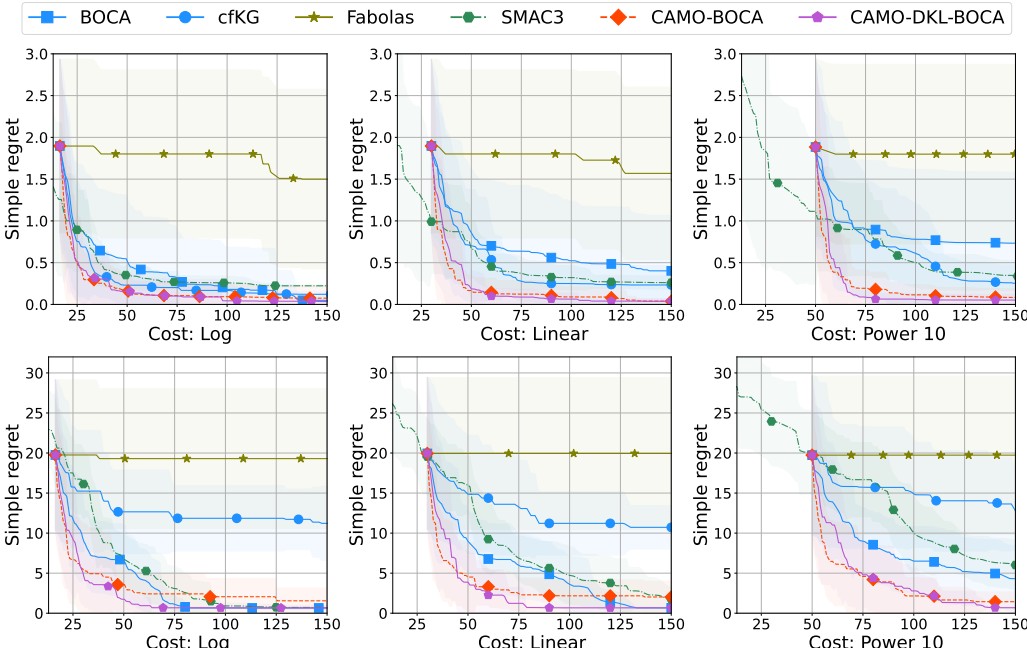

Figure 4: Continues MFBO for Currin(top row) and Bohachevsky (bottem row) function using logarithmic, linear, and exponential cost $c(t)$

**Continuous MFBO Assessment.** We show statistical SRs on the six tasks under increasing query cost in Fig. 2, which clearly demonstrates the superiority of our CAMO-BOCA (integral ARD combined with BOCA acquisition) and CAMO-DKL-BOCA over the competitor methods consistently across all tasks. Both CAMO-BOCA and CAMO-DKL-BOCA achieve similar best converging SRs in all cases. Notably CAMO-DKL-BOCA perform very well in all cases while CAMO-BOCA is more suitable for the generic continuous MFBO tasks in terms of convergence speed. We also show the actual MFBO process under different random seeds in the Appendix I to demonstrate the consistency and superiority of our methods. Methods equipped with cfKG (including our methods) are generally less competitive in terms of SRs and convergence speed compared to those with BOCA, indicating that the importance of the MFBO acquisition function and the superior of BOCA in most cases.

These MFBO contain only 1 or 2 design variables. To investigate the scalability we further evaluate it on three more complex functions: Borehole, Colville (Song et al., 2019), and Himmelblau (Dong et al., 2015), which have 6, 4, and 2 design variables, respectively. The results are shown in Fig. 3. Except that CAMO is still the best in most cases, we can see the advantage of CAMO is more significant for the high dimensional tasks.

**MFBO under different cost function** $c(t)$**.** One of the key factor in MFBO is the cost $c(t)$ of querying the high-fidelity function. To explore this, we conducted experiments with different costs, specifically three types of cost experiments: $c(t) = 10^t$, $c(t) = 5 * t$, and $c(t) = \log_2(2 + t)$ for

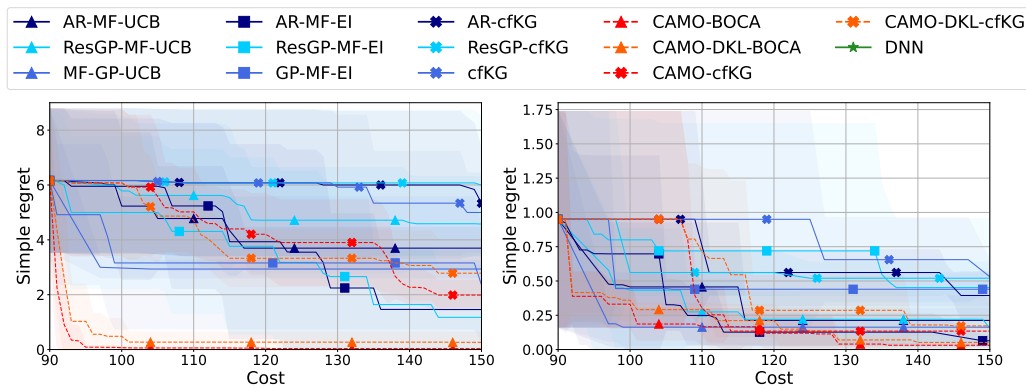

Figure 5: Discrete MFBO on Branin (left) and Currin function (right)

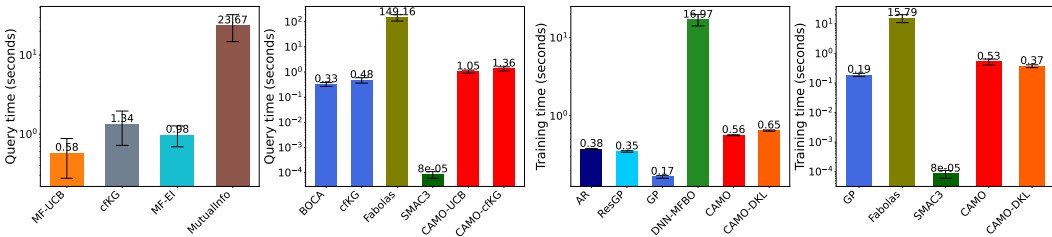

Figure 6: Acquisition optimization time on discretized and continuous MFBO (Left two figures) and training time of models on discretized and continuous MFBO (Right two figures).

the Currin and Bohachevsky function (experiments for other functions are in the Appendix I to save space). CAMO with cfKG is not shown due to the poor performance shown previously. The results are shown in Fig. 4. We can see clearly that an exponential cost setting is more challenging for all methods, as the cost increases exponentially with the fidelity index. Nonetheless, CAMO-BOCA and CAMO-DKL-BOCA outperforms all competitors in all cost settings. The advantage is more and more significant from left to right, which highlights the advantage of being convergence-aware in MFBO by maximizing the benefit/cost ratio. For a logarithmic cost setting, CAMO-BOCA and CAMO-DKL-BOCA perform similarly to BOCA because being "convergence-aware" is less "rewarded" when the cost also increases logarithmically. In contrast, if the cost increases beyond linear, the advantage of CAMO becomes significant. Such a setting is common in real-world applications, e.g., FEM (cubic complexity with mesh resolution (Hughes, 2012)) and Monte Carlo estimation (quadratic sample requirements (Caflisch, 1998)).

**Discrete MFBO Assessment.** To examine CAMO's performance on discrete MFBO, we discretize Branin and Currin into ten discrete fidelities. We also compare with the SOTA DNN-based MFBO method, DNN-MFBO (Li et al., 2020). The statistical results of SRs under increasing query costs are shown in Fig. 5. We can see that CAMO-BOCA and CAMO-DKL-BOCA outperform all counterparts by a large margin in the Branin function, while the significant advantage is less pronounced in the Currin function. However, the perform is still the best consistently across almost all query costs. In our experiment, DNN-MFBO is not shown as the results are away from the chart area, which indicates that DNN-MFBO is not competitive in this setting

**Computational Cost Analysis.** For MFBO, the time for model update and acquisition function optimization is also crucial. The average time (over all the benchmarks, iterations, and seeds) of acquisition optimization and model training is shown in Fig. 6. Despite the good reported performance in Li et al. (2020), MI and FABOLAS are impractically slow in terms of acquisition optimization time. CAMO is at the average level in terms of both acquisition optimization time and model training time, which is acceptable for practical use. SMAC3 is the fastest, but it is not competitive in terms of SRs and convergence speed in most cases as shown in previous experiments. Overall, CAMO archive a good trade-off between performance and computational cost.

## 6.2 REAL-WORLD APPLICATIONS IN ENGINEERING DESIGN

To demonstrate the practical utility of CAMO, we evaluate its performance on two real-world engineering design problems.

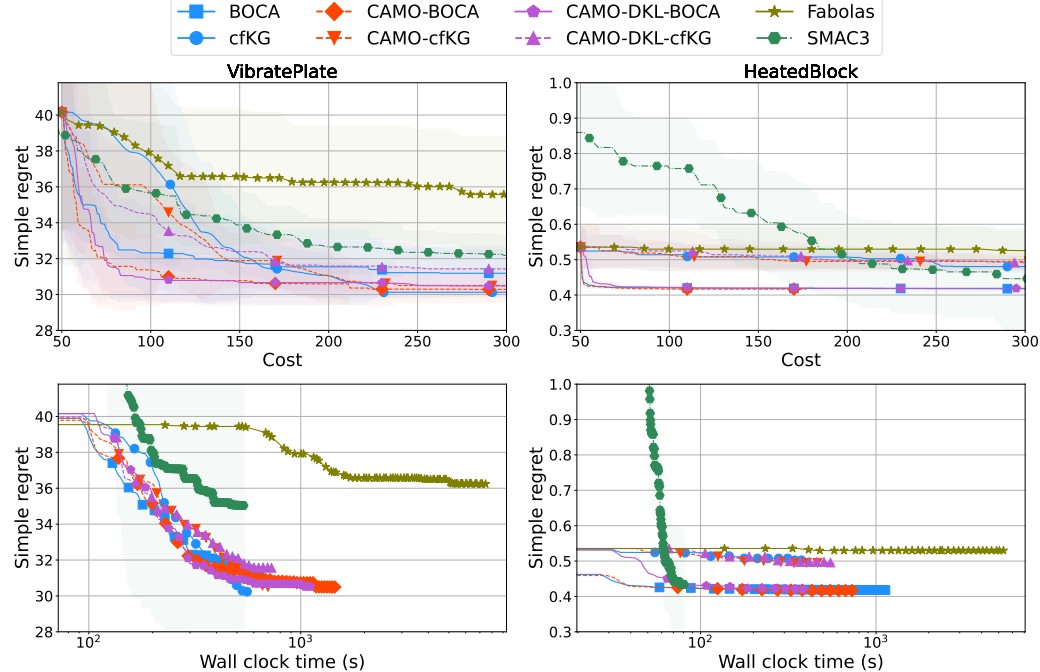

Figure 7: MFBO on Mechanical Plate Vibration (left) and Thermal Conductor Design (right) with different query cost (top row) and Wallclock time (bottom row).

**Mechanical Plate Vibration Design.** We aim to optimize three material properties, Young's modulus (in $[1 \times 1011, 5 \times 1011]$), Poisson ratio (in $[0.2, 0.6]$) and mass density (in $[6 \times 103, 9 \times 103]$), to maximize the fourth vibration mode frequency of a 3-D simply supported, square, elastic plate, of size $10 \times 10 \times 1(m)$. The frequency is calculated by solving the discretized plate using mesh-based numerical solvers. The maximum element size $H_{max} \in [1.2 - 0.2](m)$ is used to control the fidelity as a continuous variable.

**Thermal Conductor Design.** Given the property of a particular thermal conductor, our goal is to optimize the shape of the central hole where we fix the conductor to make the heat conduction (from left to right) as fast as possible. The shape of the hole (an ellipse) is described by three parameters: x-radius, y-radius, and angle. We used the time to reach 70 degrees as the objective function value, which is calculated by solving the discretized plate using mesh-based numerical solvers. The maximum element size $H_{max} \in [2 - 0.1](m)$ is used to control the fidelity as a continuous variable.

Discrete MFBO methods are not applicable to these tasks and thus are not included in the comparison. The statistical SRs versus simulation cost and wallclock time are shown in Fig. 7. We can see that CAMO-BOCA and CAMO-DKL-BOCA outperform other methods in both tasks by a large margin. CAMO-BOCA and CAMO-DKL-BOCA achieve similar best converging SRs in mechanical plate vibration design, whereas CAMO-BOCA achieve significantly faster and lower convergence SRs in the thermal conductor design. This highlight the advantage of staying simple and keep "let the data speak" rules in real-world applications using an integer ARD. In contrast, the DKL is indeed powerful but might be less effective particular when the data is scarce in optimization tasks.

## 7 CONCLUSIONS

In this work, we propose a convergence-aware MFBO by exploring MF surrogates that capture the convergence behavior of the objective function and derive two special forms for MF surrogates, a tractable integral ARD kernel and a data-driven Deep Kernel. We also derive a tighter regret bound when using the integral ARD kernel with BOCA. Our empirical evaluation shows a significant improvement with the proposed convergence-aware approach. For practical applications, we recommend using the integral ARD kernel with BOCA for continuous optimization due to its simplicity and efficiency. Both kernel will benefit from a more efficient acquisition function tailored for CAMO. CAMO may suffer from limitations such as (1) computational overhead from kernel optimization, (2) need for sufficient low-fidelity samples to learn convergence behavior, (3) current theoretical analysis limited to specific kernel forms, (4) potential challenges in hyperparameter tuning.

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

# SUPPLEMENTARY MATERIALS

## A  GAUSSIAN PROCESS

Due to its capacity to handle complex black-box functions and quantify uncertainty, the Gaussian process (GP) is often a popular selection for the surrogate model. For now, let's contemplate a simplified scenario where we possess noise-influenced observations, denoted as $y_i = f(\mathbf{x}_i) + \varepsilon, i = 1, \ldots, N$. In a GP model, a prior distribution is placed over the true underlying function of $f(\mathbf{x})$, whose input is $\mathbf{x}$:

$$\eta(\mathbf{x})|\boldsymbol{\theta} \sim \mathcal{GP}\left(m_0(\mathbf{x}), k(\mathbf{x}, \mathbf{x}'|\boldsymbol{\theta})\right), \tag{A.1}$$

employing mean and covariance functions,

$$\begin{aligned} m_0(\mathbf{x}) &= \mathbb{E}[f(\mathbf{x})], \\ k(\mathbf{x}, \mathbf{x}'|\boldsymbol{\theta}) &= \mathbb{E}[(f(\mathbf{x}) - m_0(\mathbf{x}))(f(\mathbf{x}') - m_0(\mathbf{x}'))], \end{aligned} \tag{A.2}$$

where the hyperparameters $\boldsymbol{\theta}$ control the kernel function and $\mathbb{E}[\cdot]$ is the expectation. Centering the data allows us to assume the mean function as a constant, $m_0(\mathbf{x}) \equiv m_0$. Other options, like a linear function of $\mathbf{x}$, are possible but are seldom employed unless there is prior knowledge of the function's shape. Various forms can be adopted for the covariance function, with the ARD kernel being the most widely favored.

$$k(\mathbf{x}, \mathbf{x}'|\boldsymbol{\theta}) = \theta_0 \exp\left(-(\mathbf{x} - \mathbf{x}')^T \text{diag}(\theta_1^{-2}, \ldots, \theta_l^{-2})(\mathbf{x} - \mathbf{x}')\right). \tag{A.3}$$

Starting here, we remove the specific mention of $\boldsymbol{\theta}$'s dependence on $k(x, x')$. The hyperparameters $\theta_1, \ldots, \theta_l$ are termed as length-scales in this case. When $\mathbf{x}$ is a constant parameter, $f(\mathbf{x})$ represents its random variable. In contrast, a set of values, $f(\mathbf{x}_i)$, where $i = 1, \ldots, N$, constitutes a partial realization of the GP. Realizations of GPs are deterministic functions of $x$. The key characteristic of GPs is that the joint distribution of $\eta(x_i)$, for $i = 1, \ldots, N$, is a multivariate Gaussian.

We can derive the model likelihood by assuming the Gaussian distribution of the model deficiency $\varepsilon \sim \mathcal{N}(0, \sigma^2)$, along with the utilization of the prior (A.1) and the existing data.

$$\begin{aligned} \mathcal{L} &\triangleq p(\mathbf{y}|\mathbf{x}, \boldsymbol{\theta}) = \int (f(\mathbf{x}) + \varepsilon) d\, f = \mathcal{N}(\mathbf{y}|m_0\mathbf{1}, \mathbf{K} + \sigma^2\mathbf{I}) \\ &= -\frac{1}{2}(\mathbf{y} - m_0\mathbf{1})^T (\mathbf{K} + \sigma^2\mathbf{I})^{-1}(\mathbf{y} - m_0\mathbf{1}) \\ &\quad - \frac{1}{2}\ln|\mathbf{K} + \sigma^2\mathbf{I}| - \frac{N}{2}\log(2\pi), \end{aligned} \tag{A.4}$$

where $\mathbf{K} = [K_{ij}]$ is the covariance matrix, in which $K_{ij} = k(\mathbf{x}_i, \mathbf{x}_j), i, j = 1, \ldots, N$. The hyperparameters $\boldsymbol{\theta}$ are frequently obtained through point estimations, utilizing the maximum likelihood (MLE) of Eq. (A.4) with respect to $\boldsymbol{\theta}$. The joint Gaussian distribution of $\mathbf{y}$ and $f(\mathbf{x})$ also possesses a mean value of $m_0\mathbf{1}$ and a covariance matrix.

$$\mathbf{K}' = \left[\begin{array}{c|c} \mathbf{K} + \sigma^2\mathbf{I} & \mathbf{k}(\mathbf{x}) \\ \hline \mathbf{k}^T(\mathbf{x}) & k(\mathbf{x}, \mathbf{x}) + \sigma^2 \end{array}\right], \tag{A.5}$$

where $\mathbf{k}(\mathbf{x}) = (k(\mathbf{x}_1, \mathbf{x}), \ldots, k(\mathbf{x}_N, \mathbf{x}))^T$. The conditional predictive distribution at $\mathbf{x}$ is obtained by conditioning on $\mathbf{y}$.

$$\begin{aligned} \hat{f}(\mathbf{x})|\mathbf{y} &\sim \mathcal{N}\left(\mu(\mathbf{x}), v(\mathbf{x}, \mathbf{x}')\right), \\ \mu(\mathbf{x}) &= m_0\mathbf{1} + \mathbf{k}(\mathbf{x})^T \left(\mathbf{K} + \sigma^2\mathbf{I}\right)^{-1}(\mathbf{y} - m_0\mathbf{1}), \\ v(\mathbf{x}) &= \sigma^2 + k(\mathbf{x}, \mathbf{x}) - \mathbf{k}^T(\mathbf{x})\left(\mathbf{K} + \sigma^2\mathbf{I}\right)^{-1}\mathbf{k}(\mathbf{x}). \end{aligned} \tag{A.6}$$

Given by $\mu(\mathbf{x})$, the expected value is $\mathbb{E}[f(\mathbf{x})]$, and $v(\mathbf{x})$ represents the predictive variance. The transition from Eq. (A.5) to Eq. (A.6) is critical, as the prediction posterior of this wake relies on a comparable block covariance matrix.

## B  PROOF OF LEMMA 1: CONVERGENCE CONDITIONS FOR FIDE

Let $V : \mathcal{X} \times \mathbb{R} \times [0, \infty) \to \mathbb{R}$ be a continuously differentiable Lyapunov function (which is a scalar function designed to analyze the stability and convergence properties of dynamical systems) satisfying the conditions of

Lemma 2. Consider the time derivative of $V$ along the trajectories of the system:

$$\frac{d}{dt}V(\mathbf{x}, y(\mathbf{x}, t), t) = \frac{\partial V(\mathbf{x}, y, t)}{\partial y}\frac{\partial y(\mathbf{x}, t)}{\partial t} + \frac{\partial V(\mathbf{x}, y, t)}{\partial t} = \frac{\partial V(\mathbf{x}, y, t)}{\partial y}\phi(\mathbf{x}, t, y(\mathbf{x}, t)) + \frac{\partial V(\mathbf{x}, y, t)}{\partial t}.$$
(A.7)

By the negative-definite condition, we have:

$$\frac{d}{dt}V(\mathbf{x}, y(\mathbf{x}, t), t) < 0,$$
(A.8)

for all $(\mathbf{x}, y) \neq (\mathbf{x}, y(\mathbf{x}, \infty))$ and $t \geq 0$, where $y(\mathbf{x}, \infty)$ represents the steady-state solution.

This implies that $V(\mathbf{x}, y(\mathbf{x}, t), t)$ is strictly decreasing along the system trajectories, except at the steady state. By the positive-definite property of $V$ with respect to $y$, we have:

$$V(\mathbf{x}, y(\mathbf{x}, t), t) > 0,$$
(A.9)

for all $(\mathbf{x}, y) \neq (\mathbf{x}, y(\mathbf{x}, \infty))$ and $t \geq 0$.

Since $V$ is strictly decreasing and bounded below by zero, it follows that:

$$\lim_{t \to \infty} V(\mathbf{x}, y(\mathbf{x}, t), t) = 0,$$
(A.10)

for all $\mathbf{x} \in \mathcal{X}$.

By the continuity of $V$ and the positive-definite property with respect to $y$, we conclude that:

$$\lim_{t \to \infty} y(\mathbf{x}, t) = y(\mathbf{x}, \infty),$$
(A.11)

for all $\mathbf{x} \in \mathcal{X}$, where $y(\mathbf{x}, \infty)$ represents the steady-state solution.

Therefore, under the given conditions, the solution $y(\mathbf{x}, t)$ of the general multi-fidelity system converges to the steady state $y(\mathbf{x}, \infty)$ as $t$ tends to infinity.

This revised proof corrects the previous mistake and clarifies that the solution $y(\mathbf{x}, t)$ converges to a steady state $y(\mathbf{x}, \infty)$ rather than a specific equilibrium point $y^*$. The Lyapunov function conditions ensure that the solution converges to a steady state, which can depend on the input $\mathbf{x}$.

## C   PROOF OF LEMMA 2: CONVERGENCE CONDITIONS FOR LINEAR FIDE

Lemma 1 (Convergence of Linear Multi-Fidelity System): Consider the linear multi-fidelity system given by:

$$\dot{y}(\mathbf{x}, t) = \beta(\mathbf{x}, t)y(\mathbf{x}, t) + u(\mathbf{x}, t),$$
(A.12)

where $\mathbf{x} \in \mathcal{X}$ is the input, $t \geq 0$ is the time, $y(\mathbf{x}, t)$ is the solution, $\beta(\mathbf{x}, t)$ is the system dynamics function, and $u(\mathbf{x}, t)$ is the input function.

Suppose the following conditions hold:

1. Stable Dynamics: $\beta(\mathbf{x}, t)$ is strictly negative for all $\mathbf{x} \in \mathcal{X}$ and $t \geq 0$, i.e.,

$$\beta(\mathbf{x}, t) < 0, \quad \forall \mathbf{x} \in \mathcal{X}, \forall t \geq 0.$$
(A.13)

2. Convergent Input: $u(\mathbf{x}, t)$ converges to a desired target value $u^*(\mathbf{x})$ as $t$ tends to infinity, i.e.,

$$\lim_{t \to \infty} u(\mathbf{x}, t) = u^*(\mathbf{x}), \quad \forall \mathbf{x} \in \mathcal{X}.$$
(A.14)

Then, the solution $y(\mathbf{x}, t)$ of the linear multi-fidelity system converges to a steady state $y^*(\mathbf{x})$ as $t$ tends to infinity, given by:

$$\lim_{t \to \infty} y(\mathbf{x}, t) = y^*(\mathbf{x}) = -\frac{u^*(\mathbf{x})}{\beta(\mathbf{x}, \infty)},$$
(A.15)

where $\beta(\mathbf{x}, \infty)$ denotes the limit of $\beta(\mathbf{x}, t)$ as $t$ tends to infinity.

Proof: The solution of the linear multi-fidelity system can be expressed as:

$$y(\mathbf{x}, t) = e^{\int_0^t \beta(\mathbf{x}, s)ds}y(\mathbf{x}, 0) + \int_0^t e^{\int_s^t \beta(\mathbf{x}, u)du}u(\mathbf{x}, s)ds.$$
(A.16)

Under the stable dynamics condition, the exponential term $e^{\int_0^t \beta(\mathbf{x}, s)ds}$ decays to zero as $t$ tends to infinity since $\beta(\mathbf{x}, t)$ is strictly negative. Consequently,

$$\lim_{t \to \infty} e^{\int_0^t \beta(\mathbf{x}, s)ds}y(\mathbf{x}, 0) = 0.$$
(A.17)

Under the convergent input condition, the input function $u(\mathbf{x}, t)$ converges to the target value $u^*(\mathbf{x})$ as $t$ tends to infinity. By the dominated convergence theorem, the integral term converges to:

$$\lim_{t \to \infty} \int_0^t e^{\int_s^t \beta(\mathbf{x}, u) du} u(\mathbf{x}, s) ds = -\frac{u^*(\mathbf{x})}{\beta(\mathbf{x}, \infty)}. \tag{A.18}$$

Combining the limits of the exponential term and the integral term, we obtain:

$$\lim_{t \to \infty} y(\mathbf{x}, t) = y^*(\mathbf{x}) = -\frac{u^*(\mathbf{x})}{\beta(\mathbf{x}, \infty)}. \tag{A.19}$$

Therefore, under the given conditions, the solution $y(\mathbf{x}, t)$ of the linear multi-fidelity system converges to the steady state $y^*(\mathbf{x})$ as $t$ tends to infinity.

This lemma provides a formal statement of the convergence conditions for the linear multi-fidelity system and establishes the steady-state solution under these conditions. The proof outlines the key steps in deriving the convergence result based on the properties of the system dynamics function $\beta(\mathbf{x}, t)$ and the input function $u(\mathbf{x}, t)$.

The lemma can serve as a foundation for designing and analyzing linear multi-fidelity optimization algorithms, where the functions $\beta(\mathbf{x}, t)$ and $u(\mathbf{x}, t)$ can incorporate information from different fidelity levels to guide the convergence of the solution towards the desired optimum.

# D    CLOSED-FROM SOLUTION TO LINEAR FIDE

## D.1    JOINT COVARIANCE FUNCTION FOR LINEAR FIDE

The covariance function in Equation (6) is derived by computing the covariance of the solution to the linear FiDE given the GP priors on $y(\mathbf{x}, t_0)$ and $u(\mathbf{x}, t)$. Here, we provide a detailed explanation of this derivation. We begin with the solution to the linear FiDE:

$$y(\mathbf{x}, t) = \exp\left(\int_{t_0}^t \beta(\mathbf{x}, s) ds\right) y(\mathbf{x}, t_0) + \int_{t_0}^t \exp\left(\int_s^t \beta(\mathbf{x}, u) du\right) u(\mathbf{x}, s) ds \tag{A.20}$$

Given the GP priors:

$$y(\mathbf{x}, t_0) \sim \mathcal{GP}(0, k_0(\mathbf{x}, \mathbf{x}')) \quad u(\mathbf{x}, t) \sim \mathcal{GP}(0, k_u(\mathbf{x}, t, \mathbf{x}', t')) \tag{A.21}$$

To derive the covariance function $k(\mathbf{x}, t, \mathbf{x}', t')$, we compute $\mathbb{E}[y(\mathbf{x}, t) y(\mathbf{x}', t')]$:

$$k(\mathbf{x}, t, \mathbf{x}', t') = \mathbb{E}[y(\mathbf{x}, t) y(\mathbf{x}', t')] \tag{A.22}$$

Expanding this expectation using the solution from the linear FiDE:

$$k(\mathbf{x}, t, \mathbf{x}', t') = \mathbb{E}\bigg[\bigg(\exp\bigg(\int_{t_0}^t \beta(\mathbf{x}, s) ds\bigg) y(\mathbf{x}, t_0)$$
$$+ \int_{t_0}^t \exp\bigg(\int_s^t \beta(\mathbf{x}, u) du\bigg) u(\mathbf{x}, s) ds\bigg) \cdot \exp\bigg(\int_{t_0}^{t'} \beta(\mathbf{x}', s) ds\bigg) y(\mathbf{x}', t_0) \tag{A.23}$$
$$+ \int_{t_0}^{t'} \exp\bigg(\int_s^{t'} \beta(\mathbf{x}', u) du\bigg) u(\mathbf{x}', s) ds\bigg]$$

Expanding this product and using the linearity of expectation, we get four terms. The cross-terms involving both $y(\mathbf{x}, t_0)$ and $u(\mathbf{x}, t)$ vanish due to their independence and zero mean. The remaining two terms give us Equation (6):

$$k(\mathbf{x}, t, \mathbf{x}', t') = \exp\bigg(\int_{t_0}^t \beta(\mathbf{x}, s) ds\bigg) \exp\bigg(\int_{t_0}^{t'} \beta(\mathbf{x}', s) ds\bigg) k_0(\mathbf{x}, \mathbf{x}')$$
$$+ \int_{t_0}^t \int_{t_0}^{t'} \exp\bigg(\int_s^t \beta(\mathbf{x}, u) du\bigg) \exp\bigg(\int_{s'}^{t'} \beta(\mathbf{x}', u) du\bigg) k_u(\mathbf{x}, s, \mathbf{x}', s') ds ds' \tag{A.24}$$

This derivation shares similarities with the work of Alvarez et al. (2013) on linear latent force models, where they also derive covariance functions for dynamic systems modeled by linear differential equations. While their focus was on modeling dynamic systems, our application extends this approach to multi-fidelity optimization.

## D.2 POLYNOMIAL $\beta(t)$ AND ARD KERNEL

The following derivation are based on a separable kernel, i.e., $k^u(\mathbf{x}, t, \mathbf{x}', t') = k_x^u(\mathbf{x}, \mathbf{x}')k_t^u(t, t')$. For cases where the kernel is not separable, we can use the spectral representation and decompose the kernel into a sum of separable kernels.

When $\beta(t)$ is a polynomial of order $P$, we can derive a closed-form solution for the covariance function. Let's consider a polynomial of the form:

$$\beta(t) = \sum_{p=0}^{P} a_p t^p, \tag{A.25}$$

where $a_p$ are the coefficients of the polynomial. Given the ARD kernel for $k_t^u(t, t')$:

$$k_t^u(t, t') = \sigma^2 \exp\left(-\frac{(t - t')^2}{2\ell^2}\right), \tag{A.26}$$

and the polynomial $\beta(t)$, the covariance function of the solution becomes:

$$k(\mathbf{x}, t, \mathbf{x}', t') = \exp\left(\sum_{p=0}^{P} \frac{a_p}{p+1}(t^{p+1} - t_0^{p+1})\right) \exp\left(\sum_{p=0}^{P} \frac{a_p}{p+1}(t'^{p+1} - t_0^{p+1})\right) k^0(\mathbf{x}, \mathbf{x}')$$

$$+ \sigma^2 k_{\mathbf{x}}^u(\mathbf{x}, \mathbf{x}') \int_{t_0}^{t} \int_{t_0}^{t'} \exp\left(\sum_{p=0}^{P} \frac{a_p}{p+1}(t^{p+1} - s^{p+1})\right) \exp\left(\sum_{p=0}^{P} \frac{a_p}{p+1}(t'^{p+1} - s'^{p+1})\right) \exp\left(-\frac{(s - s')^2}{2\ell^2}\right) ds\, ds'. \tag{A.27}$$

The double integral in the second term can be evaluated analytically using the properties of the Gaussian integral and the exponential integral. After some algebraic manipulations, we obtain:

$$k(\mathbf{x}, t, \mathbf{x}', t') = \exp\left(\sum_{p=0}^{P} \frac{a_p}{p+1}(t^{p+1} - t_0^{p+1})\right) \exp\left(\sum_{p=0}^{P} \frac{a_p}{p+1}(t'^{p+1} - t_0^{p+1})\right) k^0(\mathbf{x}, \mathbf{x}')$$

$$+ \sigma^2 k_{\mathbf{x}}^u(\mathbf{x}, \mathbf{x}') \sqrt{\frac{\pi \ell^2}{2}} \exp\left(\sum_{p=0}^{P} \frac{a_p}{p+1}(t^{p+1} + t'^{p+1} - 2t_0^{p+1})\right)$$

$$\times \sum_{n=0}^{\infty} \frac{1}{n!} \left(\frac{\ell^2}{2}\right)^n \sum_{k=0}^{n} \binom{n}{k}(-1)^k \prod_{p=0}^{P} \left(\sum_{m=0}^{p} \frac{a_m}{m+1}\left(t^{m+1-p+k} - t_0^{m+1-p+k}\right)\right)$$

$$\times \prod_{p=0}^{P} \left(\sum_{m=0}^{p} \frac{a_m}{m+1}\left(t'^{m+1-p+n-k} - t_0^{m+1-p+n-k}\right)\right). \tag{A.28}$$

This closed-form expression for the covariance function provides insights into the behavior of the GP-based LiFiDE model with a polynomial $\beta(t)$. The covariance function depends on the spatial covariance $k_{\mathbf{x}}^u(\mathbf{x}, \mathbf{x}')$, the initial condition covariance $k^0(\mathbf{x}, \mathbf{x}')$, the polynomial coefficients $a_p$, the signal variance $\sigma^2$, and the length scale $\ell$ of the ARD kernel.

The closed-form solution involves an infinite sum over the index $n$, which can be approximated by truncating the sum to a finite number of terms based on the desired accuracy and computational constraints. The number of terms needed for a good approximation depends on the values of the polynomial coefficients and the length scale of the ARD kernel.

The polynomial representation of $\beta(t)$ allows for flexibility in capturing various time-dependent behaviors of the multi-fidelity system. The order of the polynomial, $P$, determines the complexity of the temporal patterns that can be modeled. Higher-order polynomials can capture more intricate temporal dynamics but also introduce more parameters to be learned.

The closed-form solution enables efficient computation and can be used for inference and learning tasks in the GP-based LiFiDE model with a polynomial $\beta(t)$. It also facilitates the interpretation of the model parameters and their effects on the covariance structure.

## D.3 CONSTANT $\beta(t) = \beta$ AND PERIODIC KERNEL

The analytical results for the covariance function can be extended to periodic kernels as well. Let's consider the case where $k_t^u(t, t')$ is a periodic kernel, which captures periodic patterns in the temporal domain. A commonly

used periodic kernel is the exponential sine squared (ESS) kernel, defined as:

$$k_t^u(t, t') = \sigma^2 \exp\left(-\frac{2\sin^2(\pi|t - t'|/p)}{\ell^2}\right),$$

(A.29)

where $\sigma^2$ is the signal variance, $\ell$ is the length scale, and $p$ is the period of the periodic pattern.

Assuming a constant $\beta(t) = \beta$, the covariance function of the solution with the ESS kernel becomes:

$$\begin{aligned} k(\mathbf{x}, t, \mathbf{x}', t') =& e^{\beta(t-t_0)} e^{\beta(t'-t_0)} k^0(\mathbf{x}, \mathbf{x}') \\ &+ \sigma^2 k_\mathbf{x}^u(\mathbf{x}, \mathbf{x}') \int_{t_0}^t \int_{t_0}^{t'} e^{\beta(t-s)} e^{\beta(t'-s')} \exp\left(-\frac{2\sin^2(\pi|s - s'|/p)}{\ell^2}\right) ds ds'. \end{aligned}$$

(A.30)

The double integral in the second term can be evaluated analytically using the properties of the exponential sine squared function and the exponential integral and we obtain:

$$\begin{aligned} k(\mathbf{x}, t, \mathbf{x}', t') =& e^{\beta(t-t_0)} e^{\beta(t'-t_0)} k^0(\mathbf{x}, \mathbf{x}') \\ &+ \sigma^2 k_\mathbf{x}^u(\mathbf{x}, \mathbf{x}') \left(\frac{p}{2\pi} \sum_{n=-\infty}^{\infty} \frac{1}{\beta - \frac{2\pi in}{p}} \left[e^{\beta(t-t_0)-\frac{2\pi in}{p}(t-t_0)} - e^{\beta(t'-t_0)-\frac{2\pi in}{p}(t'-t_0)}\right] \times \exp\left(-\frac{2}{\ell^2}\right) B_n\left(\frac{2}{\ell^2}\right)\right), \end{aligned}$$

(A.31)

where $B_n(\cdot)$ is the modified Bessel function of the first kind of order $n$.

This analytical result for the covariance function with the ESS kernel provides insights into the behavior of the GP-based LiFiDE model when dealing with periodic temporal patterns. The covariance function depends on the spatial covariance $k_\mathbf{x}^u(\mathbf{x}, \mathbf{x}')$, the initial condition covariance $k^0(\mathbf{x}, \mathbf{x}')$, the constant $\beta$, the signal variance $\sigma^2$, the length scale $\ell$, and the period $p$ of the periodic kernel. The analytical expression involves an infinite sum over the index $n$, which can be approximated by truncating the sum to a finite number of terms based on the desired accuracy and computational constraints.

The ESS kernel captures periodic patterns in the fidelity domain, allowing the GP-based FiDE model to handle multi-fidelity systems with periodic or oscillating behaviors. The period $p$ determines the frequency of the periodic pattern, while the length scale $\ell$ controls the smoothness of the periodic variations.

### D.4 CONSTANT $\beta(t) = \beta$ AND MATÉRN KERNEL

The Matérn kernel is a generalization of the squared exponential kernel and provides more flexibility in modeling the smoothness of the underlying function.

The Matérn kernel with parameter $\nu$ is defined as:

$$k_t^u(t, t') = \sigma^2 \frac{2^{1-\nu}}{\Gamma(\nu)} \left(\sqrt{2\nu} \frac{|t - t'|}{\ell}\right)^\nu K_\nu\left(\sqrt{2\nu} \frac{|t - t'|}{\ell}\right),$$

(A.32)

where $\sigma^2$ is the signal variance, $\ell$ is the characteristic length scale, $\Gamma(\cdot)$ is the gamma function, and $K_\nu(\cdot)$ is the modified Bessel function of the second kind of order $\nu$.

For $\nu = \frac{3}{2}$, the Matérn kernel is given by:

$$k_t^u(t, t') = \sigma^2 \left(1 + \frac{\sqrt{3}|t - t'|}{\ell}\right) \exp\left(-\frac{\sqrt{3}|t - t'|}{\ell}\right).$$

(A.33)

The corresponding covariance function of the solution becomes:

$$\begin{aligned} k(\mathbf{x}, t, \mathbf{x}', t') =& e^{\beta(t-t_0)} e^{\beta(t'-t_0)} k^0(\mathbf{x}, \mathbf{x}') \\ &+ \sigma^2 k_\mathbf{x}^u(\mathbf{x}, \mathbf{x}') \int_{t_0}^t \int_{t_0}^{t'} e^{\beta(t-s)} e^{\beta(t'-s')} \left(1 + \frac{\sqrt{3}|s - s'|}{\ell}\right) \exp\left(-\frac{\sqrt{3}|s - s'|}{\ell}\right) ds ds'. \end{aligned}$$

(A.34)

The double integral in the second term can be evaluated analytically, yielding:

$$k(\mathbf{x}, t, \mathbf{x}', t') = e^{\beta(t-t_0)} e^{\beta(t'-t_0)} k^0(\mathbf{x}, \mathbf{x}')$$

$$+ \sigma^2 k_{\mathbf{x}}^u(\mathbf{x}, \mathbf{x}') \Bigg( \frac{1}{\beta^2 - \frac{3}{\ell^2}} \left[ e^{\beta(t-t_0)} + e^{\beta(t'-t_0)} - e^{-\frac{\sqrt{3}}{\ell}|t-t'|} \left( e^{\beta(t-t_0) - \frac{\sqrt{3}}{\ell}(t'-t_0)} + e^{\beta(t'-t_0) - \frac{\sqrt{3}}{\ell}(t-t_0)} \right) \right]$$

$$+ \frac{\sqrt{3}}{\ell} \left[ \frac{1}{\beta^2 - \frac{3}{\ell^2}} \left( e^{\beta(t-t_0)} + e^{\beta(t'-t_0)} \right) - \frac{1}{\beta + \frac{\sqrt{3}}{\ell}} \left( e^{\beta(t-t_0) - \frac{\sqrt{3}}{\ell}(t-t_0)} + e^{\beta(t'-t_0) - \frac{\sqrt{3}}{\ell}(t'-t_0)} \right) \right]$$

$$- \frac{1}{\beta + \frac{\sqrt{3}}{\ell}} \left[ e^{\beta(t-t_0) - \frac{\sqrt{3}}{\ell}(t-t_0)} + e^{\beta(t'-t_0) - \frac{\sqrt{3}}{\ell}(t'-t_0)} \right]$$

$$+ \frac{1}{\beta - \frac{\sqrt{3}}{\ell}} \left[ e^{\beta(t-t_0) + \frac{\sqrt{3}}{\ell}(t-t_0)} + e^{\beta(t'-t_0) + \frac{\sqrt{3}}{\ell}(t'-t_0)} \right] \Bigg).$$

$$(A.35)$$

For $\nu = \frac{5}{2}$, the Matérn kernel is given by:

$$k_t^u(t, t') = \sigma^2 \left( 1 + \frac{\sqrt{5}|t - t'|}{\ell} + \frac{5(t - t')^2}{3\ell^2} \right) \exp\left( -\frac{\sqrt{5}|t - t'|}{\ell} \right). \tag{A.36}$$

The corresponding covariance function of the solution becomes:

$$k(\mathbf{x}, t, \mathbf{x}', t') = e^{\beta(t-t_0)} e^{\beta(t'-t_0)} k^0(\mathbf{x}, \mathbf{x}')$$

$$+ \sigma^2 k_{\mathbf{x}}^u(\mathbf{x}, \mathbf{x}') \int_{t_0}^{t} \int_{t_0}^{t'} e^{\beta(t-s)} e^{\beta(t'-s')} \left( 1 + \frac{\sqrt{5}|s - s'|}{\ell} + \frac{5(s - s')^2}{3\ell^2} \right) \exp\left( -\frac{\sqrt{5}|s - s'|}{\ell} \right) ds \, ds'.$$

$$(A.37)$$

The double integral in the second term can be evaluated analytically, yielding:

$$k(\mathbf{x}, t, \mathbf{x}', t') = e^{\beta(t-t_0)} e^{\beta(t'-t_0)} k^0(\mathbf{x}, \mathbf{x}')$$

$$+ \sigma^2 k_{\mathbf{x}}^u(\mathbf{x}, \mathbf{x}') \Bigg( \frac{1}{\beta^2 - \frac{5}{\ell^2}} \left[ e^{\beta(t-t_0)} + e^{\beta(t'-t_0)} - e^{-\frac{\sqrt{5}}{\ell}|t-t'|} \left( e^{\beta(t-t_0) - \frac{\sqrt{5}}{\ell}(t'-t_0)} + e^{\beta(t'-t_0) - \frac{\sqrt{5}}{\ell}(t-t_0)} \right) \right]$$

$$+ \frac{\sqrt{5}}{\ell} \left[ \frac{1}{\beta^2 - \frac{5}{\ell^2}} \left( e^{\beta(t-t_0)} + e^{\beta(t'-t_0)} \right) - \frac{1}{\beta + \frac{\sqrt{5}}{\ell}} \left( e^{\beta(t-t_0) - \frac{\sqrt{5}}{\ell}(t-t_0)} + e^{\beta(t'-t_0) - \frac{\sqrt{5}}{\ell}(t'-t_0)} \right) \right]$$

$$+ \frac{5}{3\ell^2} \left[ \frac{1}{\beta^2 - \frac{5}{\ell^2}} \left( e^{\beta(t-t_0)} + e^{\beta(t'-t_0)} \right) - \frac{1}{\beta + \frac{\sqrt{5}}{\ell}} \left( e^{\beta(t-t_0) - \frac{\sqrt{5}}{\ell}(t-t_0)} + e^{\beta(t'-t_0) - \frac{\sqrt{5}}{\ell}(t'-t_0)} \right) \right]$$

$$- \frac{1}{\beta + \frac{\sqrt{5}}{\ell}} \left[ e^{\beta(t-t_0) - \frac{\sqrt{5}}{\ell}(t-t_0)} + e^{\beta(t'-t_0) - \frac{\sqrt{5}}{\ell}(t'-t_0)} \right]$$

$$+ \frac{1}{\beta - \frac{\sqrt{5}}{\ell}} \left[ e^{\beta(t-t_0) + \frac{\sqrt{5}}{\ell}(t-t_0)} + e^{\beta(t'-t_0) + \frac{\sqrt{5}}{\ell}(t'-t_0)} \right]$$

$$- \frac{5}{3\ell^2} \left[ \frac{1}{\beta^2 - \frac{5}{\ell^2}} \left( e^{\beta(t-t_0)} + e^{\beta(t'-t_0)} \right) - \frac{1}{(\beta + \frac{\sqrt{5}}{\ell})^2} \left( e^{\beta(t-t_0) - \frac{\sqrt{5}}{\ell}(t-t_0)} + e^{\beta(t'-t_0) - \frac{\sqrt{5}}{\ell}(t'-t_0)} \right) \right]$$

$$+ \frac{5}{3\ell^2} \left[ \frac{1}{(\beta - \frac{\sqrt{5}}{\ell})^2} \left( e^{\beta(t-t_0) + \frac{\sqrt{5}}{\ell}(t-t_0)} + e^{\beta(t'-t_0) + \frac{\sqrt{5}}{\ell}(t'-t_0)} \right) \right] \Bigg).$$

$$(A.38)$$

These closed-form solutions for the Matérn kernel with $\nu = \frac{3}{2}$ and $\nu = \frac{5}{2}$ allow for efficient computation of the covariance function without the need for numerical integration. The Matérn kernel provides additional flexibility in modeling the smoothness of the underlying function compared to the squared exponential kernel, while still maintaining analytical tractability when combined with the constant $\beta(t) = \beta$ assumption.

It is important to note that the choice of the Matérn kernel parameter $\nu$ depends on the prior knowledge or assumptions about the smoothness of the underlying function. Higher values of $\nu$ correspond to smoother functions, while lower values allow for more rough or irregular behavior. The closed-form solutions presented here cover two commonly used cases, but the approach can be extended to other values of $\nu$ if needed.

### D.5 CONSTANT $\beta(t) = \beta$ AND RANDOM FOURIER FEATURES

Since SE, periodic, and Matérn kernels are all stationary kernels, we can try to extend the closed-form solution to any stationary kernel. Recall that any stationary kernel can be represented as a Fourier series, and the random Fourier features (RFF) kernel defined as:

$$k_t^u(t, t') = \frac{\sigma^2}{m} \sum_{i=1}^{m} \cos(\omega_i(t - t')), \tag{A.39}$$

where $\sigma^2$ is the signal variance, $m$ is the number of random features, and $\omega_i$ are randomly sampled frequencies from a distribution $p(\omega)$ (typically a Gaussian distribution with zero mean and variance $1/\ell^2$, where $\ell$ is the length scale of the squared exponential kernel being approximated).

Assuming a constant $\beta(t) = \beta$, the covariance function of the solution with the RFF kernel becomes:

$$
\begin{aligned}
k(\mathbf{x}, t, \mathbf{x}', t') = & e^{\beta(t-t_0)} e^{\beta(t'-t_0)} k^0(\mathbf{x}, \mathbf{x}') \\
& + \frac{\sigma^2}{m} k_{\mathbf{x}}^u(\mathbf{x}, \mathbf{x}') \sum_{i=1}^{m} \int_{t_0}^{t} \int_{t_0}^{t'} e^{\beta(t-s)} e^{\beta(t'-s')} \cos(\omega_i(s - s')) ds ds'.
\end{aligned}
\tag{A.40}
$$

The double integral in the second term can be evaluated analytically for each frequency $\omega_i$. After some algebraic manipulations, we obtain:

$$
\begin{aligned}
k(\mathbf{x}, t, \mathbf{x}', t') = & e^{\beta(t-t_0)} e^{\beta(t'-t_0)} k^0(\mathbf{x}, \mathbf{x}') \\
& + \frac{\sigma^2}{m} k_{\mathbf{x}}^u(\mathbf{x}, \mathbf{x}') \sum_{i=1}^{m} \left( \frac{1}{\beta^2 + \omega_i^2} \Big[ e^{\beta(t-t_0)} \cos(\omega_i(t - t_0)) + e^{\beta(t'-t_0)} \cos(\omega_i(t' - t_0)) \right. \\
& \left. - e^{\beta(t+t'-2t_0)} \cos(\omega_i(t - t')) \Big] \right. \\
& \left. + \frac{\omega_i}{\beta^2 + \omega_i^2} \Big[ e^{\beta(t-t_0)} \sin(\omega_i(t - t_0)) - e^{\beta(t'-t_0)} \sin(\omega_i(t' - t_0)) \Big] \right).
\end{aligned}
\tag{A.41}
$$

This analytical result for the covariance function with the RFF kernel provides a tractable solution for linear FiDE. The covariance function depends on the spatial covariance $k_{\mathbf{x}}^u(\mathbf{x}, \mathbf{x}')$, the initial condition covariance $k^0(\mathbf{x}, \mathbf{x}')$, the constant $\beta$, the signal variance $\sigma^2$, and the randomly sampled frequencies $\omega_i$.

The RFF kernel approximates the squared exponential kernel by projecting the input data onto a set of random Fourier features. The number of random features $m$ controls the quality of the approximation, with larger values of $m$ leading to more accurate approximations but also increasing the computational cost.

### D.6 EXTENSION TO POLYNOMIAL $\beta(t)$ AND RANDOM FOURIER FEATURES

We aim to generalize the closed-form solution to the case when $\beta(t)$ is a polynomial of $p$-th order and $k_t^u(t, t')$ is a random Fourier kernel such that it can almost cover all the cases with stationary kernels. Let's consider a polynomial $\beta(t)$ of order $p$:

$$\beta(t) = \sum_{i=0}^{p} a_i t^i, \tag{A.42}$$

and the random Fourier kernel:

$$k_t^u(t, t') = \frac{\sigma^2}{M} \sum_{m=1}^{M} \cos(\omega_m(t - t') + b_m), \tag{A.43}$$

where $a_i$ are the polynomial coefficients, $\omega_m$ are the random Fourier frequencies drawn from a probability distribution (e.g., Gaussian distribution), $b_m$ are random phase shifts drawn uniformly from $[0, 2\pi]$, and $M$ is the number of random features.

The covariance function of the solution becomes:

$$
\begin{aligned}
k(\mathbf{x}, t, \mathbf{x}', t') = & \exp\left( \sum_{i=0}^{p} \frac{a_i}{i+1}(t^{i+1} - t_0^{i+1}) \right) \exp\left( \sum_{i=0}^{p} \frac{a_i}{i+1}(t'^{i+1} - t_0^{i+1}) \right) k^0(\mathbf{x}, \mathbf{x}') \\
& + \frac{\sigma^2}{M} k_{\mathbf{x}}^u(\mathbf{x}, \mathbf{x}') \sum_{m=1}^{M} \int_{t_0}^{t} \int_{t_0}^{t'} \exp\left( \sum_{i=0}^{p} \frac{a_i}{i+1}(t^{i+1} - s^{i+1}) \right) \exp\left( \sum_{i=0}^{p} \frac{a_i}{i+1}(t'^{i+1} - s'^{i+1}) \right) \cos(\omega_m(s - s') + b_m) ds ds'.
\end{aligned}
\tag{A.44}
$$

To evaluate the double integral in the second term, we can use the properties of the cosine function and the exponential function. The integral can be split into two parts:

$$I_m = \int_{t_0}^{t} \int_{t_0}^{t'} \exp\left(\sum_{i=0}^{p} \frac{a_i}{i+1}(t^{i+1} - s^{i+1})\right) \exp\left(\sum_{i=0}^{p} \frac{a_i}{i+1}(t'^{i+1} - s'^{i+1})\right) \cos(\omega_m(s - s') + b_m) ds ds'$$

$$= \cos(b_m)I_{m,c} - \sin(b_m)I_{m,s},$$

(A.45)

where $I_{m,c}$ and $I_{m,s}$ are the integrals corresponding to the cosine and sine terms, respectively.

The integrals $I_{m,c}$ and $I_{m,s}$ can be evaluated using integration by parts and the properties of the exponential and trigonometric functions. The resulting expressions will involve terms of the form $\exp(\cdot)$, $\sin(\cdot)$, and $\cos(\cdot)$, with arguments depending on the polynomial coefficients $a_i$, the time points $t$, $t'$, $t_0$, and the random Fourier frequencies $\omega_m$.

The final expression for the covariance function will be:

$$k(\mathbf{x}, t, \mathbf{x}', t') = \exp\left(\sum_{i=0}^{p} \frac{a_i}{i+1}(t^{i+1} - t_0^{i+1})\right) \exp\left(\sum_{i=0}^{p} \frac{a_i}{i+1}(t'^{i+1} - t_0^{i+1})\right) k^0(\mathbf{x}, \mathbf{x}')$$

$$+ \frac{\sigma^2}{M} k_{\mathbf{x}}^u(\mathbf{x}, \mathbf{x}') \sum_{m=1}^{M} \left[\cos(b_m)I_{m,c} - \sin(b_m)I_{m,s}\right],$$

(A.46)

where $I_{m,c}$ and $I_{m,s}$ are the evaluated integrals.

The resulting expression for the covariance function will be more complex compared to the previous cases, as it will involve higher-order terms and more intricate expressions for the integrals. The complexity of the expression will depend on the order $p$ of the polynomial $\beta(t)$.

Although the final expression may not have a simple closed form, it can still be computed numerically for given values of the polynomial coefficients, time points, and random Fourier frequencies and phase shifts. The numerical computation will involve evaluating the integrals $I_{m,c}$ and $I_{m,s}$ for each random feature $m$ and summing over all the features.

The linear FiDE with a polynomial $\beta(t)$ and a random Fourier kernel can still be used for inference and learning tasks, but the computational complexity will be higher compared to the other case. The increased complexity arises from the higher-order terms in the polynomial and the need to evaluate the integrals numerically. In practice, the choice of the polynomial order $p$ will depend on the complexity of the time-dependent behavior being modeled and the available computational resources. Higher-order polynomials can capture more complex temporal patterns but also increase the computational burden.

# E   THEORETICAL ANALYSIS

## E.1   CONVERGENCE BEHAVIOR OF THE INTEGRAL ARD KERNEL

For the integral ARD kernel in Eq. (8),

$$k(\mathbf{x}, t, \mathbf{x}', t') = e^{\beta(t-t_0)}e^{\beta(t'-t_0)}k^0(\mathbf{x}, \mathbf{x}') + k_{\mathbf{x}}^u(\mathbf{x}, \mathbf{x}')I(t, t'),$$

(A.47)

let $k^0(\mathbf{x}, \mathbf{x}')$ be zero as we consider large $t$ where the decay of the kernel is dominated by the integral term. The kernel can be simplified to I(t,t') which indicate the connection between fidelity levels. Rewrite the kernel in a more compact form:

$$I(t, t') = \sqrt{\frac{\pi}{2}} \ell e^{\beta(t+t'-2t_0)} \left[\text{erf}\left(\frac{t - t_0}{\sqrt{2}\ell}\right) + \text{erf}\left(\frac{t' - t_0}{\sqrt{2}\ell}\right)\right]$$

$$- \ell^2 e^{\beta(t+t'-2t_0) + \frac{\beta^2\ell^2}{2}} \left[e^{-\beta(t-t_0)}\text{erf}\left(\frac{t - t_0 - \beta\ell^2}{\sqrt{2}\ell}\right) + e^{-\beta(t'-t_0)}\text{erf}\left(\frac{t' - t_0 - \beta\ell^2}{\sqrt{2}\ell}\right)\right].$$

(A.48)

Now, focus on the behavior of the kernel when $t$ and $t'$ are close to the maximum fidelity $T$. In this case, we can approximate the error functions using their Taylor series expansions around $\frac{t-t_0}{\sqrt{2}\ell}$ and $\frac{T-t_0-\beta\ell^2}{\sqrt{2}\ell}$, respectively.

For the first term, we have:

$$\text{erf}\left(\frac{t - t_0}{\sqrt{2}\ell}\right) \approx \text{erf}\left(\frac{T - t_0}{\sqrt{2}\ell}\right) + \frac{2}{\sqrt{\pi}}e^{-\left(\frac{T-t_0}{\sqrt{2}\ell}\right)^2}\frac{t - T}{\sqrt{2}\ell}.$$

(A.49)

For the second term, we have:

$$\text{erf}\left(\frac{t - t_0 - \beta \ell^2}{\sqrt{2}\ell}\right) \approx \text{erf}\left(\frac{T - t_0 - \beta \ell^2}{\sqrt{2}\ell}\right) + \frac{2}{\sqrt{\pi}} e^{-\left(\frac{T - t_0 - \beta \ell^2}{\sqrt{2}\ell}\right)^2} \frac{t - T}{\sqrt{2}\ell}. \tag{A.50}$$

Substituting these approximations into the kernel and simplifying, we get:

$$
\begin{aligned}
I(t, t') \approx & \sqrt{2\pi}\ell e^{\beta(T + T - 2t_0)} \text{erf}\left(\frac{T - t_0}{\sqrt{2}\ell}\right) + 2e^{\beta(T + T - 2t_0) - \left(\frac{T - t_0}{\sqrt{2}\ell}\right)^2}(t + t' - 2t) \\
& - \sqrt{2\pi}\ell^3 \beta e^{\beta(t_+ T - 2t_0) + \frac{\beta^2 \ell^2}{2}} \text{erf}\left(\frac{T - t_0 - \beta \ell^2}{\sqrt{2}\ell}\right) \\
& - 2\ell^2 e^{\beta(T + T - 2t_0) + \frac{\beta^2 \ell^2}{2} - \left(\frac{T - t_0 - \beta \ell^2}{\sqrt{2}\ell}\right)^2}(t + t' - 2t_*).
\end{aligned}
\tag{A.51}
$$

From this approximation, we can see that the kernel behaves like a squared exponential kernel with a length scale that depends on $t$, $t_0$, $\beta$, and $\ell$. To illustrate the non-stationary kernel, we visualize it in Fig. **??** with different length scales. We can see that the length scale controls how the fidelity is correlated. Particularly, when the fidelity increases, the correlation is always higher. When the length scale is large enough, the low-fidelity correlations are weak, which confines many real applications where the low-fidelity has a weak correlation due to the lack of stableness or convergence, whereas high-fidelity shows a strong correlation with a convergent trend.

The effective length near the maximum fidelity $t$ is given by:

$$\ell_{\text{eff}}^2 = \frac{1}{2}\left(\frac{T - t_0}{\sqrt{2}\ell}\right)^2 - \frac{\beta^2 \ell^2}{2} + \left(\frac{T - t_0 - \beta \ell^2}{\sqrt{2}\ell}\right)^2. \tag{A.52}$$

As $t$ increases (i.e., as we approach the maximum fidelity), the effective lengthscale $\ell_{\text{eff}}$ also increases, provided that $\beta \ell^2 < t - t_0$. This condition ensures that the second term in the expression for $\ell_{\text{eff}}^2$ doesn't dominate. It's important to note that the approximations used here are valid only when $t$ and $t'$ are close to $T$. A more rigorous analysis would be needed to derive precise regret bounds for the BOCA algorithm with this specific kernel. Nonetheless, this example illustrates how the properties of the non-stationary kernel can influence the behavior of the algorithm and the resulting regret bounds.

### E.2 BOCA Regret Bound with Integral ARD Kernel

From the previous analysis, we found that the effective lengthscale of the kernel near the maximum fidelity $T$ is given by:

$$h_{\text{eff}}^2 = \frac{1}{2}\left(\frac{T - t_0}{\sqrt{2}\ell}\right)^2 - \frac{\beta^2 \ell^2}{2} + \left(\frac{T - t_0 - \beta \ell^2}{\sqrt{2}\ell}\right)^2. \tag{A.53}$$

Let's introduce a non-stationary kernel that captures this behavior:

$$k(t, t') = \exp\left(-\frac{(t - t')^2}{2h(t)^2}\right), \tag{A.54}$$

where $h(t)$ is a function that depends on the fidelity $t$ and has the following form:

$$h(t)^2 = h_0^2 + \gamma(T - t)^{-\alpha}, \tag{A.55}$$

with $h_0$, $\gamma$, and $\alpha$ being positive constants. Now, let's compute $\xi(\tau)$ using this non-stationary kernel:

$$\xi(\tau) = \inf_{t \in \mathcal{T}} \exp\left(-\frac{(T - t)^2}{2h(t)^2}\right) \quad \text{s.t.} \quad |T - t| \le \tau. \tag{A.56}$$

Assuming that $T - \tau \ge t_0$ and $\beta \ell^2 \ll \tau$, we can approximate $\xi(\tau)$ as:

$$\xi(\tau) \approx \exp\left(-\frac{\tau^2}{2(h_0^2 + \gamma \tau^{-\alpha})}\right). \tag{A.57}$$

When $\tau$ is small, i.e., when we are close to the maximum fidelity $T$, the term $\gamma\tau^{-\alpha}$ dominates, and we have:

$$\xi(\tau) \approx \exp\left(-\frac{\tau^{2+\alpha}}{2\gamma}\right). \tag{A.58}$$

Substituting this approximation into the expression for $\mathcal{X}_\rho$:

$$\mathcal{X}_\rho \approx \left\{x \in \mathcal{X} : f_* - f(x) \leq 2\rho\sqrt{\kappa_0}\exp\left(-\frac{p^{1+\alpha/2}}{2\gamma^{1/2}}\right)\right\}. \tag{A.59}$$

The regret bound for the BOCA algorithm with this non-stationary kernel will have the following form:

$$S(\Lambda) \lesssim \sqrt{\frac{\Psi_{n_\Lambda}(\mathcal{X}_\rho)}{n_\Lambda}} + \sqrt{\frac{\Psi_{n_\Lambda}(\mathcal{X})}{n_\Lambda^{1-\alpha/2}}}, \tag{A.60}$$

where $\Psi_{n_\Lambda}(\mathcal{X}_\rho)$ and $\Psi_{n_\Lambda}(\mathcal{X})$ are the maximum information gains for the non-stationary kernel over the sets $\mathcal{X}_\rho$ and $\mathcal{X}$, respectively.

Due to the faster decay of $\xi(\tau)$ near the maximum fidelity, $\Psi_{n_\Lambda}(\mathcal{X}_\rho)$ will be smaller compared to the squared exponential kernel case, leading to a tighter first term in the regret bound. The second term in the regret bound also has a smaller exponent $1 - \alpha/2$, which further improves the bound.

In summary, by introducing a non-stationary kernel that captures the increasing lengthscale behavior near the maximum fidelity, we have derived a tighter regret bound for the BOCA algorithm. The specific values of the constants $h_0$, $\gamma$, and $\alpha$ will depend on the properties of the given kernel and the problem setup, but the general form of the regret bound remains the same. Note that this derivation relies on some assumptions and approximations, and a more rigorous analysis would be needed to obtain precise bounds. However, this example demonstrates how the insights gained from analyzing the given kernel can be used to derive tighter regret bounds for the BOCA algorithm.

# F    EFFICIENT TRAINING AND INFERENCE THROUGH SUBSET DECOMPOSITION IN AUTOREGRESSIVE GPS

The section is mainly taken from CAR Xing et al. (2024) and the reader is referred to the original paper for more details. Here we briefly show how to decompose the joint likelihood of the AR kernel to enable efficient training and inference in the multi-fidelity setting.

Given a set of observations $\mathbf{Y} = [\mathbf{Y}^{(0)}; \mathbf{Y}^{(1)}; \ldots; \mathbf{Y}^{(T)}]$, we have the joint distribution

$$\begin{pmatrix} \vec{\mathbf{y}}^{(0)} \\ \vdots \\ \vec{\mathbf{y}}^{(T)} \end{pmatrix} \sim \mathcal{N}\left(\mathbf{0}, \begin{pmatrix} \mathbf{K}^{(00)} & \cdots & \mathbf{K}^{(0T)} \\ \vdots & \ddots & \vdots \\ \mathbf{K}^{(T0)} & \cdots & \mathbf{K}^{(TT)} \end{pmatrix}\right), \tag{A.61}$$

where $\vec{\mathbf{y}}^{(0)} = \text{vec}\left(\mathbf{Y}^{(0)}\right)$ is the vectorization; $[\mathbf{K}^{(kl)}]_{ij} = k^y\left(\mathbf{x}_i, t_k, \mathbf{x}_j, t_l\right)$ is the shorthand notation of output. For a small number of total training data of all fidelity, we can simply opt for a maximum likelihood estimation (MLE) for the joint likelihood

$$\mathcal{L} = -\frac{1}{2}\vec{\mathbf{y}}^\top \mathbf{\Sigma}^{-1}\vec{\mathbf{y}} - \frac{1}{2}\log|\mathbf{\Sigma}| - \frac{ND}{2}\log(2\pi), \tag{A.62}$$

where $\mathbf{\Sigma}$ is the whole covariance matrix in Eq. (A.61) and $\vec{\mathbf{y}} = [\mathbf{y}^{(0)}, \ldots, \mathbf{y}^{(T)}]^\top$. However, this approach will soon become invalid, particularly in a multi-fidelity scenario where we expect many low-fidelity simulations.

Since the integration of covariance can be done by parts, $\mathbf{K}^{(kl)}$ admits an additive structure exactly as in AR. We can follow Gratiet and Cannamela (2015) to decompose the joint likelihood Eq. (A.62) into independent components provided that corresponding inputs strictly follow a subset structure, i.e., $\mathbf{X}^T \subset \cdots \subset \mathbf{X}^2 \subset \mathbf{X}^1$. For problems with only a small number of fidelity (a small $T$), the subset structure may not be too difficult to satisfy. However, for the infinite (countable) fidelity setting, such a requirement is not practical. Here, we derive a decomposition by introducing virtual observations $\hat{\mathbf{Y}}$ for each fidelity such that $\mathbf{Y}^{(T)}$ satisfies the subset requirement for the completed set $\{\mathbf{Y}^{(T-1)}, \hat{\mathbf{Y}}^{(T-1)}\}$. $\check{\mathbf{Y}}^{(T)}$ is the part of $\mathbf{Y}^{(T)}$ that forms the subset of $\mathbf{Y}^{(T-1)}$ (with a selection formulation $\check{\mathbf{X}}^{(T)} = \mathbf{E}^{(T)}\mathbf{X}^{(T-1)}$, where $\mathbf{X}^{(T-1)}$ corresponds to the previous-fidelity

outputs $\mathbf{Y}^{(T-1)}$). The joint likelihood can then be decomposed,

$$\mathcal{L} = \log p(\mathbf{Y}^{(0:T)}) = \log p(\mathbf{Y}^{(0:T-1)}, \mathbf{Y}^{(T)})$$

$$= \log p(\mathbf{Y}^{(0:T-1)}) + \log \int \left[ p(\mathbf{Y}^{(T)} | \mathbf{Y}^{(T-1)}, \hat{\mathbf{Y}}^{(T-1)}) \ p(\hat{\mathbf{Y}}^{(T-1)} | \mathbf{Y}^{(T-1)}) \right] d\hat{\mathbf{Y}}^{(T-1)} \tag{A.63}$$

$$= \log p(\mathbf{Y}^{(0:T-1)}) - \frac{DN^{(T)}}{2} \log(2\pi) - \frac{D}{2} \log \left| \tilde{\mathbf{K}}_a^{(TT)} \right| - \frac{1}{2} \left[ (\mathbf{Y}_a^{(T)})^\top \left( \tilde{\mathbf{K}}_a^{(TT)} \right)^{-1} \mathbf{Y}_a^{(T)} \right],$$

where
$$\tilde{\mathbf{K}}_a^{(TT)} = \mathbf{K}_a^{(TT)} + \hat{\mathbf{E}}^{(T)} \hat{\mathbf{\Sigma}}^{(T)} (\hat{\mathbf{E}}^{(T)})^\top, \tag{A.64}$$

is the updated additive kernel with the uncertainty of the predictive variance $\hat{\mathbf{\Sigma}}^{(T)}$ of the virtual points, whose computation details are given later in Section F.1;

$$[\mathbf{K}_a^{(TT)}]_{ij} = [\mathbf{K}^{(TT)}]_{ij} - [\mathbf{K}^{(T-1,T-1)}]_{ij} \tag{A.65}$$

is the additive/residual part of the kernel from $(T-1)$ to $T$;

$$\mathbf{Y}_a^{(T)} = \begin{pmatrix} \check{\mathbf{Y}}^{(T)} \\ \hat{\mathbf{Y}}^{(T)} \end{pmatrix} - e^{-\beta\Delta_T} \times \begin{pmatrix} \check{\mathbf{Y}}^{(T-1)} \\ \bar{\mathbf{Y}}^{(T-1)} \end{pmatrix} \tag{A.66}$$

is the additive part for the completed outputs from $(T-1)$ to $T$; $\Delta_T$ is the time interval between $t_{T-1}$ and $t_T$; $\log p(\mathbf{Y}^{(0:T-1)})$ is the log likelihood for the previous $T$ fidelities, which is obtained by calling Eq. (A.63) recursively. Through the decomposition of Eq. (A.63), the computation complexity is reduced from $\mathcal{O}((D \sum_{i=0}^{T} N^{t_i})^3)$ to $\mathcal{O}(D \sum_{i=1}^{T} (N^{t_i} + N^{t_{i-1}} - |\mathbf{X}^{(t_i)} \cap \mathbf{X}^{(t_{i-1})}|_n)^3)$. Here, $|\cdot|_n$ indicates the number of samples. Furthermore, when data at a new fidelity is obtained, we only need to modify the joint likelihood slightly by adding new blocks, which will be handy for active learning or Bayesian optimization. Model training is conducted easily by maximizing the joint likelihood Eq. (A.63) with respect to the hyperparameters using gradient-based optimizations.

## F.1 Predictive Posterior

Since the joint model (A.61) is a Gaussian, the predictive posterior for the highest fidelity can be derived as in standard GP with a large covariance matrix that requires inversion for once. Similar to the derivation of an efficient joint likelihood in the previous section, we drive an efficient predictive posterior $\mathbf{y}(\mathbf{x}_*, T) \sim \mathcal{N}(\bar{\mathbf{y}}(\mathbf{x}_*, T), \mathbf{\Sigma}(\mathbf{x}_*, T))$,

$$\bar{\mathbf{y}}(\mathbf{x}_*, T) = e^{-\beta\Delta_T} \times \bar{\mathbf{y}}(\mathbf{x}_*, T-1) + (\mathbf{k}_{a*}^{(TT)})^\top \left( \mathbf{K}_a^{(TT)} \right)^{-1} \mathbf{Y}_a^{(T)}$$

$$\mathbf{\Sigma}(\mathbf{x}_*, T) = e^{-2\beta\Delta_T} \times \mathbf{\Sigma}(\mathbf{x}_*, T-1) + \tilde{\mathbf{\Sigma}}^{(T)} + \mathbf{\Gamma}^{(T)} \hat{\mathbf{\Sigma}}^{(T)} (\mathbf{\Gamma}^{(T)})^\top \tag{A.67}$$

where

$$\tilde{\mathbf{\Sigma}}^{(t_T)} = \mathbf{I} \left( \tilde{\mathbf{k}}_{a*}^{(TT)} \right)^\top \left( \tilde{\mathbf{k}}_a^{(TT)} \right)^{-1} \tilde{\mathbf{k}}_{a*}^{(TT)}, \quad \hat{\mathbf{\Sigma}}^{(t_T)} = \mathbf{I} \left( \hat{\mathbf{k}}_{a*}^{(TT)} \right)^\top \left( \hat{\mathbf{K}}_a^{(TT)} \right)^{-1} \hat{\mathbf{k}}_{a*}^{(TT)},$$

$$\mathbf{\Gamma}^{(t_T)} = \left[ \mathbf{k}_{a*}^{(T,T)} \left( \mathbf{E}_n^{(T)} \right)^\top \tilde{\mathbf{k}}_{a*}^{(T-1,T-1)} \left( \tilde{\mathbf{K}}_a^{(T-1,T-1)} \right)^{-1} \right] \mathbf{E}_m^{(T)}, \tag{A.68}$$

with two selection matrixes that follow:

$$\hat{\mathbf{X}}^{(t_T)} = \left( \mathbf{E}_m^{(T)} \right)^\top [\mathbf{X}^{(T-1)}, \hat{\mathbf{X}}^{(T)}], \quad \mathbf{X}^{(t_T)} = \left( \mathbf{E}_n^{(T)} \right)^\top [\mathbf{X}^{(T-1)}, \hat{\mathbf{X}}^{(T)}]. \tag{A.69}$$

In these equations, $[\tilde{k}_{a*}^{(TT)}]_j = k^y(\mathbf{x}_*, T, \mathbf{x}_j, T) - k^y(\mathbf{x}_*, T-1, \mathbf{x}_j, T-1)$ for $\mathbf{x}_j \in \{\mathbf{X}^{(t_T)}\}$ is the kernel additional part between $T$ and $T-1$ for $\mathbf{x}_*$ and $\mathbf{X}^{(t_T)}$; $\tilde{\mathbf{\Sigma}}^{(t_T)}$ is the predictive variance based on $\{\mathbf{X}^{(T-1)}, \hat{\mathbf{X}}^{(T)}\}$ of $T$ fidelity model; $[\hat{k}_{a*}^{(TT)}]_j = k^y(\mathbf{x}_*, T, \mathbf{x}_j, T) - k^y(\mathbf{x}_*, T-1, \mathbf{x}_j, T-1)$ for $\mathbf{x}_j \in \{\hat{\mathbf{X}}^{(t_T)}\}$ is the kernel additional part between between $T$ and $T-1$ for $\mathbf{x}_*$ and $\hat{\mathbf{X}}^{(t_T)}$; $\hat{\mathbf{\Sigma}}^{(t_T)}$ is the predictive variance based on $\{\hat{\mathbf{X}}^{(T)}\}$.

# G Synthetic Benchmarks

In the experiments, we used five synthetic benchmark tasks to evaluate our method. The definitions of the objective functions are given as follows.

### G.1 PARK FUNCTION

The input is four dimensional, $\mathbf{x} = [x_1, x_2] \in [0, 1]^2$. We have one fidelity indicator, $t \in [0, 1]$, to query the function, which are involved in the function, given by:

$$f(x, t) = \frac{(x_1 + 0.5t)^2 + (x_2 + 0.5t)^2}{2}. \tag{A.70}$$

### G.2 CURRIN FUNCTION

The input is four dimensional, $\mathbf{x} = [x_1, x_2] \in [0, 1]^2$. We have one fidelity indicator, $t \in [0, 1]$, to query the function, which are involved in the function, given by:

$$f(x, t) = [1 - \exp(-\frac{1}{2x_2 t})]\frac{2300x_1^3 + 1900x_1^2 + 2092x_1 + 60}{100x_1^3 + 500x_1^2 + 4x_1 + 20}. \tag{A.71}$$

### G.3 BRANIN FUNCTION

The input is two dimensional, $\mathbf{x} = [x_1, x_2] \in [0, 1.5] \times [0, 1.5]$. We have continuous fidelity indicator, $t \in [0, 1]$, to query the function, which are involved in the function, given by

$$f(x, t) = [x_2 - [b - 0.1 \times (1 - t)]x_1^2 + cx_1 - r]^2 + 10(1 - s)\cos(x_1) + 10, \tag{A.72}$$

where $b = \frac{5.1}{4\pi^2}$, $c = \frac{5}{\pi}$, $r = 6$ and $s = \frac{1}{8\pi}$.

### G.4 NONLINEAR SIN FUNCTION

Two-level multi-fidelity function where input is one dimension, $\mathbf{x} \in [0, 1.5]$ and fidelity indicator $t \in [0, 1]$, where low and high fidelity is given by:

$$\begin{aligned} f_{\text{low}}(x) &= \sin(8\pi x), \\ f_{\text{high}}(x) &= (x - \sqrt{2})f_{\text{low}}(x)^2. \end{aligned} \tag{A.73}$$

The visualization of the low and high fidelity functions is shown in Figure 8, and the global maximum is 0.0 at $x = 1.5$.

### G.5 FORRESTER FUNCTION

Two-level multi-fidelity function where input is one dimension, $\mathbf{x} \in [0, 1.5]$ and fidelity indicator $t \in [0, 1]$, where low and high fidelity is given by:

$$\begin{aligned} f_{\text{low}}(x) &= 0.5f_{\text{high}}(x) + 10(x - 0.5) + 5, \\ f_{\text{high}}(x) &= (6x - 2)^2 \sin(12x - 4). \end{aligned} \tag{A.74}$$

The visualization of the low and high fidelity functions is shown in Figure 8.

### G.6 BOHACHEVSKY FUNCTION

Two-level multi-fidelity function where input is two dimension, $\mathbf{x} \in [-5, 5]$ and fidelity indicator $t \in [-5, 5]$, where low and high fidelity is given by:

$$\begin{aligned} f_h(x_1, x_2) &= x_1^2 + 2x_2^2 - 0.3\cos(3\pi x_1) - 0.4\cos(4\pi x_2) + 0.7, \\ f_l(x_1, x_2) &= f_h(0.7x_1, x_2) + x_1 x_2 - 12 \end{aligned} \tag{A.75}$$

### G.7 BOREHOLE FUNCTION

Two-level multi-fidelity function where input is six dimension, $r_w \in [0.05, 0.15], r \in [100, 50000], T_u \in [63070, 115600], H_u \in [990, 1110], T_l \in [63.1, 116], H_l \in [700, 820], L \in [1120, 1680], K_w \in$

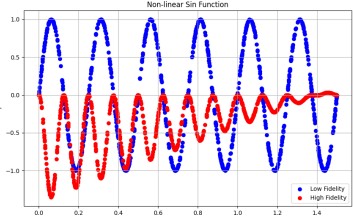
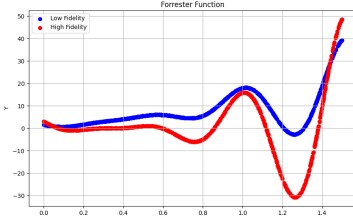

Figure 8: Low and high fidelity NonLinearSin function (Left) and Forrester function (Right) within input range.

$[9855, 12045]$, where low and high fidelity is given by:

$$f_b(x, A, B) = \frac{A \cdot T_u \cdot (H_u - H_l)}{\log\left(\dfrac{r}{r_w}\right) \cdot \left(B + \dfrac{2L \cdot T_u}{\log\left(\dfrac{r}{r_w}\right) \cdot r_w^2 \cdot K_w} + \dfrac{T_u}{T_l}\right)} \quad \text{(A.76)}$$

$$f_h(x) = f_b(x, 2\pi, 1)$$
$$f_l(x) = f_b(x, 5, 1.5)$$

### G.8 COLVILLE FUNCTION

Two-level multi-fidelity function where input is four dimension, $\mathbf{x} \in [-1, 1]$ and coefficient $A <= 0.68$, where low and high fidelity is given by:

$$f_h = 100 * (x_1^2 - x_2)^2 + (x_1 - 1)^2 + (x_3 - 1)^2$$
$$+90 * (x_3^2 - x_4) + 10.1 * ((x_2 - 1)^2 + (x_4 - 1)^2) + 19.8 * (x_2 - 1) * (x_4 - 1), \quad \text{(A.77)}$$
$$f_l = f_{high} * (A^2(x_1, x_2, x_3, x_4)) - (A + 0.5)(5 * x_1^2 + 4 * x_2^2 + 3 * x_3^2 + x_4^2)$$

### G.9 HIMMELBLAU FUNCTION

Two-level multi-fidelity function where input is two dimension, $\mathbf{x} \in [-1, 1]$ and coefficient $A <= 0.68$, where low and high fidelity is given by:

$$f_h(x_1, x_2) = (x_1^2 + x_2 - 11)^2 + (x_2^2 + x_1 - 7)^2,$$
$$f_l(x_1, x_2) = f_h(0.5x_1, 0.8x_2) + x_2^3 - (x_1 + 1)^2 \quad \text{(A.78)}$$

## H DETAILS OF REAL-WORLD APPLICATIONS

### H.1 MECHANICAL PLATE VIBRATION DESIGN

The objective of this application is to optimize the design of a 3-D simply supported, square, elastic plate with dimensions $10 \times 10 \times 1$, as illustrated in Fig. 9. The primary goal is to identify materials that maximize the fourth vibration mode frequency, thereby minimizing the risk of resonance-induced damage caused by interactions with other components. The material properties under consideration include Young's modulus (ranging from $1 \times 10^{11}$ to $5 \times 10^{11}$), Poisson's ratio (between 0.2 and 0.6), and mass density (varying from $6 \times 10^3$ to $9 \times 10^3$). To accurately compute the vibration frequency, the plate is discretized using quadratic tetrahedral elements, as depicted in Fig. 9. Two levels of fidelity are considered in this analysis.

### H.2 THERMAL CONDUCTOR DESIGN

The second application focuses on optimizing the design of a thermal conductor, as shown in Fig. 10a. The heat source is located on the left side of the conductor, with the temperature initially at zero and increasing to 100 degrees within 0.5 seconds. The heat transfer occurs through the conductor towards the right end. The conductor's dimensions and material properties, including thermal conductivity and mass density (both equal to 1), are fixed. To facilitate installation, a hole must be bored in the center of the conductor. The top, bottom, and inner surfaces of the hole are thermally insulated, preventing heat transfer across these boundaries. The size

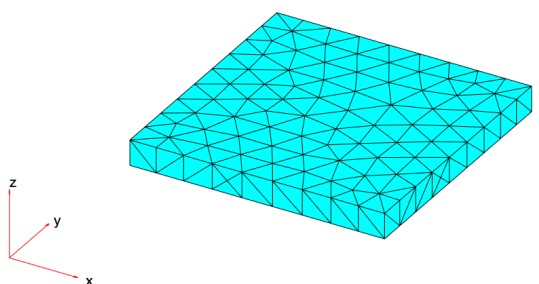

Figure 9: Quadratic tetrahedral element discretization of the plate ($H_{max} = 1.2$).

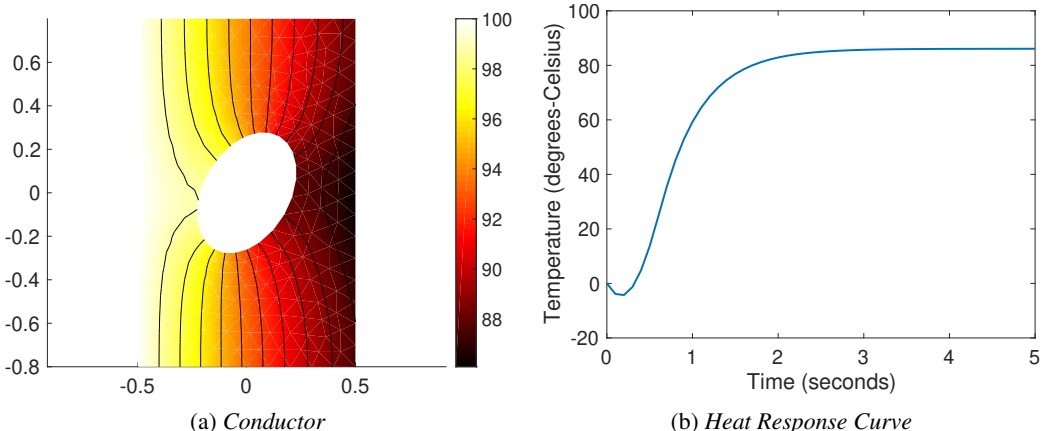

(a) *Conductor*                                    (b) *Heat Response Curve*

Figure 10: The thermal conductor with one transient heat solution (a), and the heat responsive curve on the right edge (b). The white triangles in (a) are the finite elements used to discretize the conductor to compute the solution.

and angle of the hole play a crucial role in determining the speed of heat transfusion. In general, the hole is an ellipse, characterized by three parameters: x-radius, y-radius, and angle. The objective is to minimize the time required for the temperature at the right end to reach 70 degrees, thus maximizing the heat conduction rate from left to right.

To evaluate the time taken to reach the target temperature, the conductor is discretized using quadratic tetrahedral elements. The finite element method is then applied to solve a transient heat transfer problem, yielding a response heat curve at the right edge, as exemplified in Fig. 10b. By analyzing this response curve, the time at which the temperature reaches 70 degrees can be determined.

# I  ADDITIONAL EXPERIMENTAL RESULTS

To explore the performance of different models combined with different acquisition functions, we provide more detailed information on experiments with different costs in Fig. 11. The conclusion is consistent with the main text. Basically, CAMO is the best choice particularly when the cost function is an exponential-like function, which confronts real-world applications such as finite element analysis and neural network architecture search (NAS).

Below we provide more detailed information on the sampling functions for all continuous MFBO experiments conducted in Section 6 Each marker represents a sampling point. Among the numerous seeds, we selected the first four for display. The results for Branin, Currin, Park, Nonlinear sin, Forrester, Bohachevsky, Borehole, Colville, and Himmelblau are shown in Fig. 12, Fig. 13, Fig. 14, Fig. 15, Fig. 16, Fig. 17, Fig. 18, Fig. 19, and Fig. 21, respectively.

We can see the advantage of CAMO in almost all cases in terms of the query cost and the simple regret. The query cost is significantly reduced compared to the other methods, and the simple regret is also competitive. particularly, CAMO always shows a very fast convergence rate in the early stage of the optimization process. Taking a closer look, CAMO-BOCA conducts most of its experience in the low-fidelity region, which is consistent with the theoretical analysis in order to reduce the query cost.

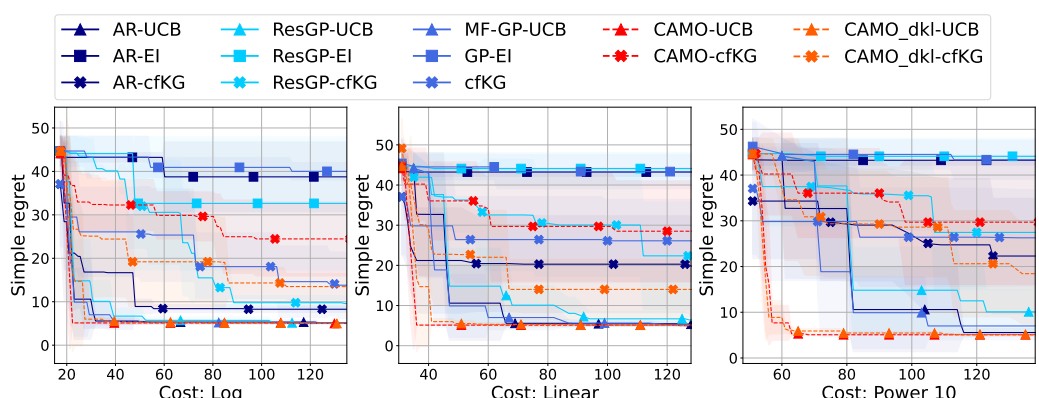

Figure 11: MFBO for Forrester function with different cost.

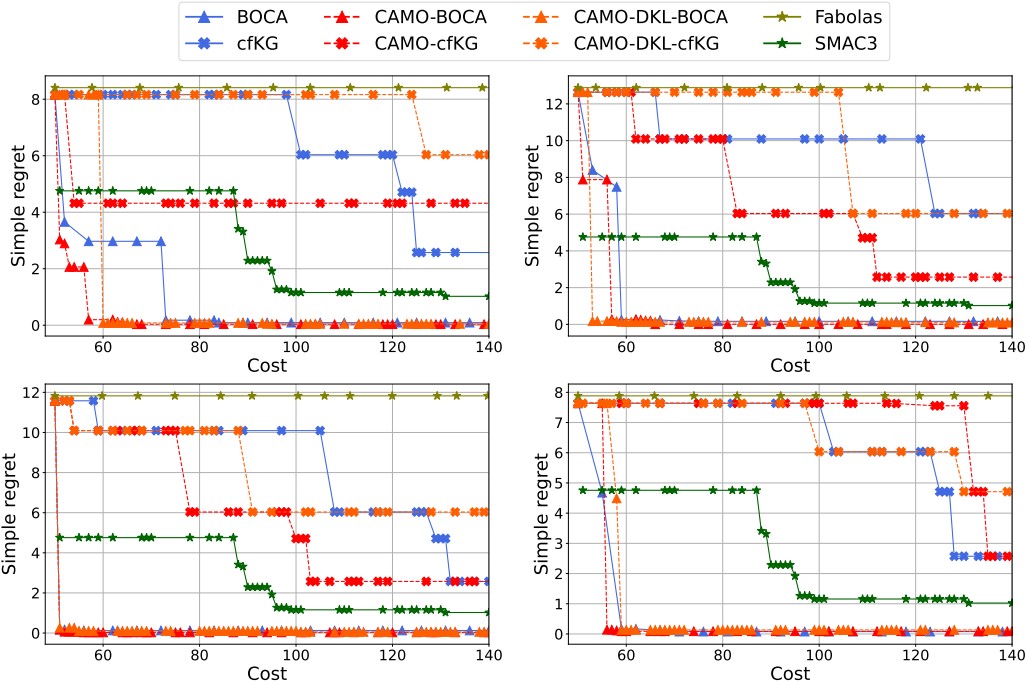

Figure 12: MFBO results from four seeds on Branin function.

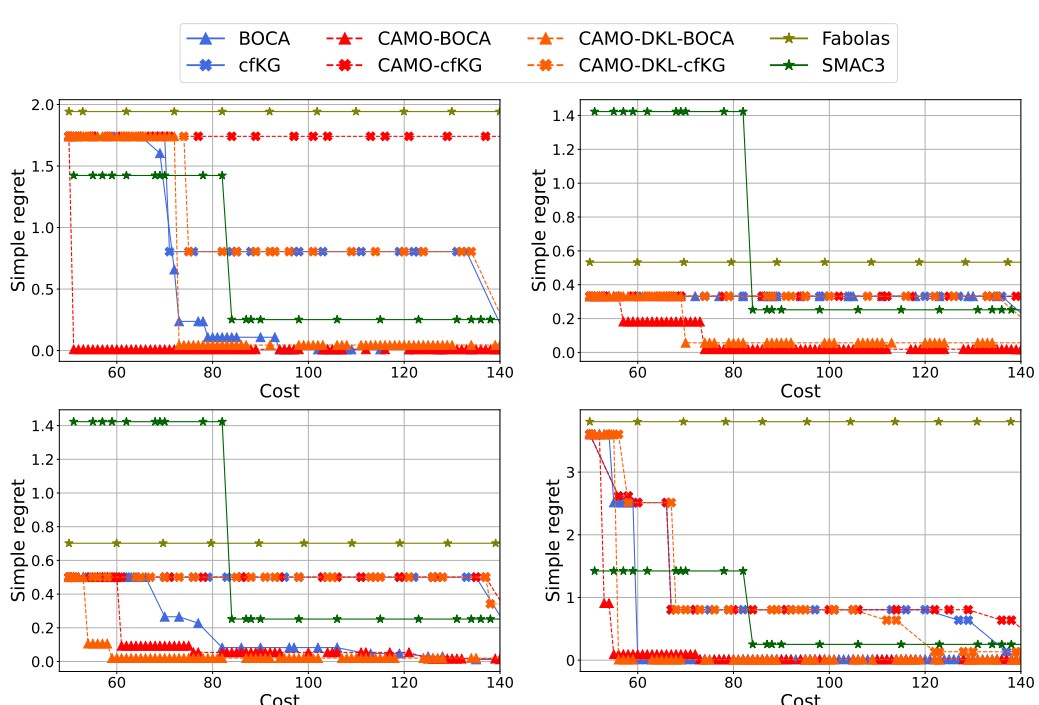

Figure 13: MFBO results from four seeds on Currin function.

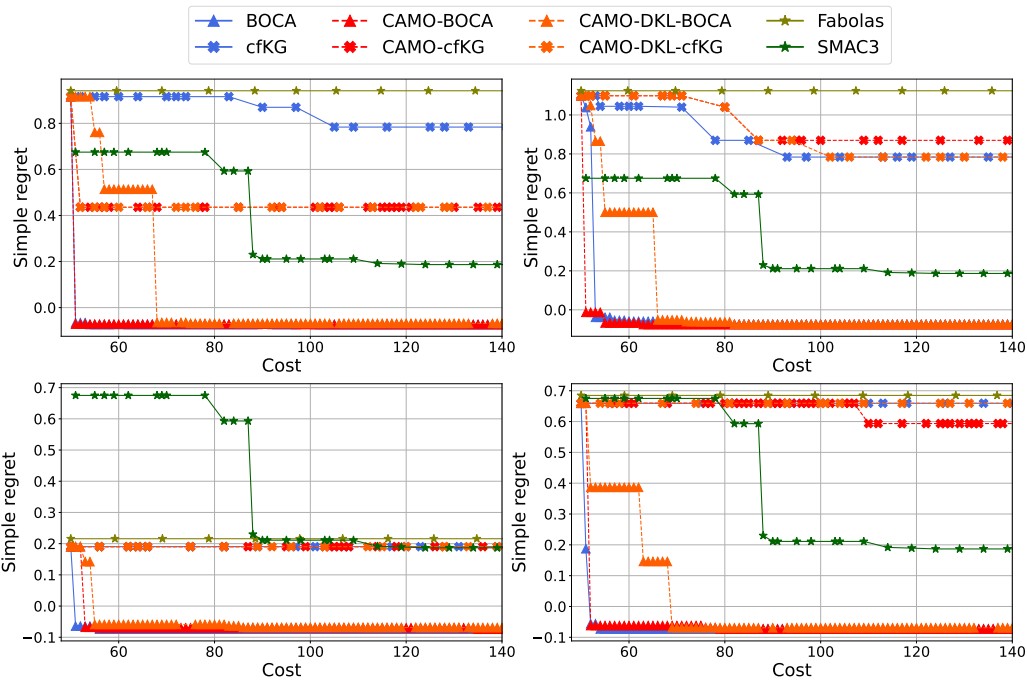

Figure 14: MFBO results from four seeds on Park function.

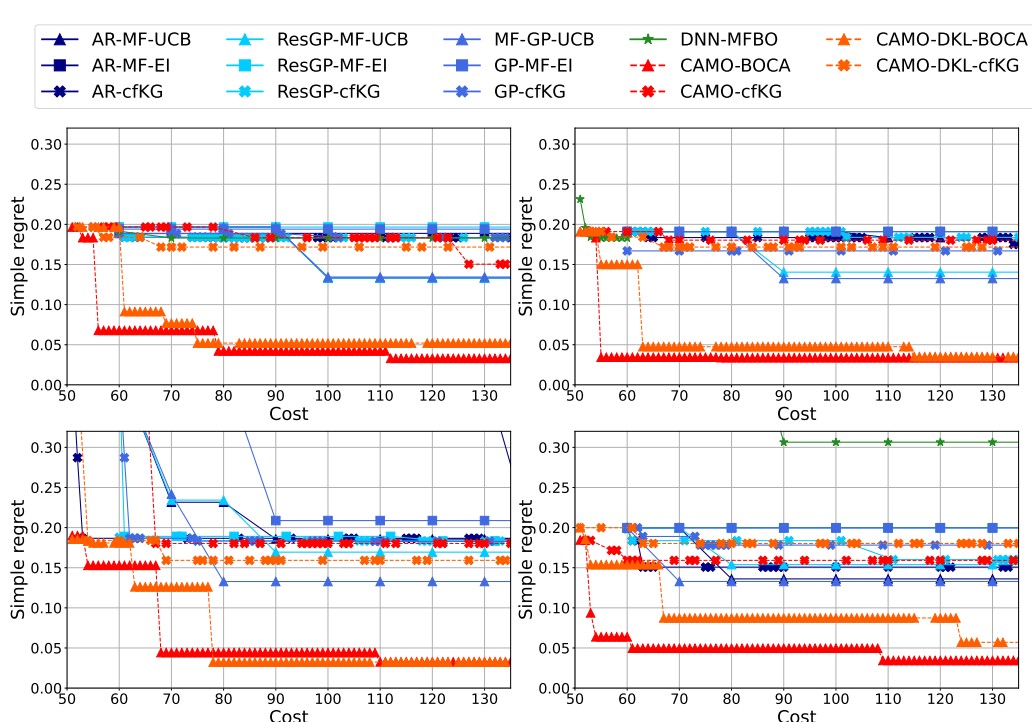

Figure 15: MFBO results from four seeds on Nonlinear sin function.

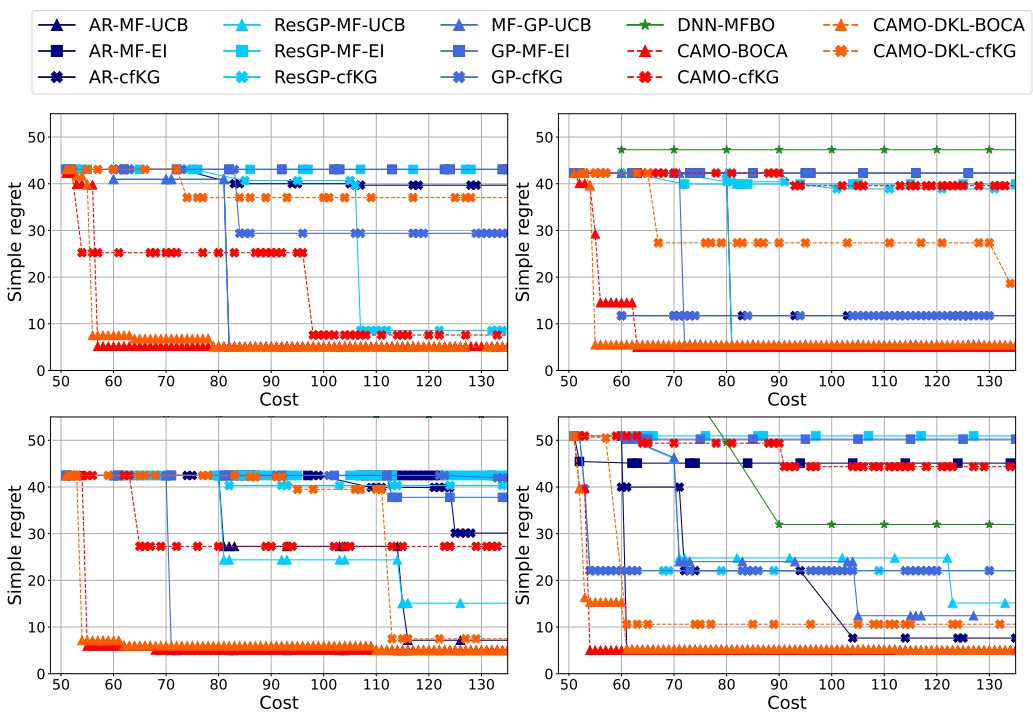

Figure 16: MFBO results from four seeds on Forrester function.

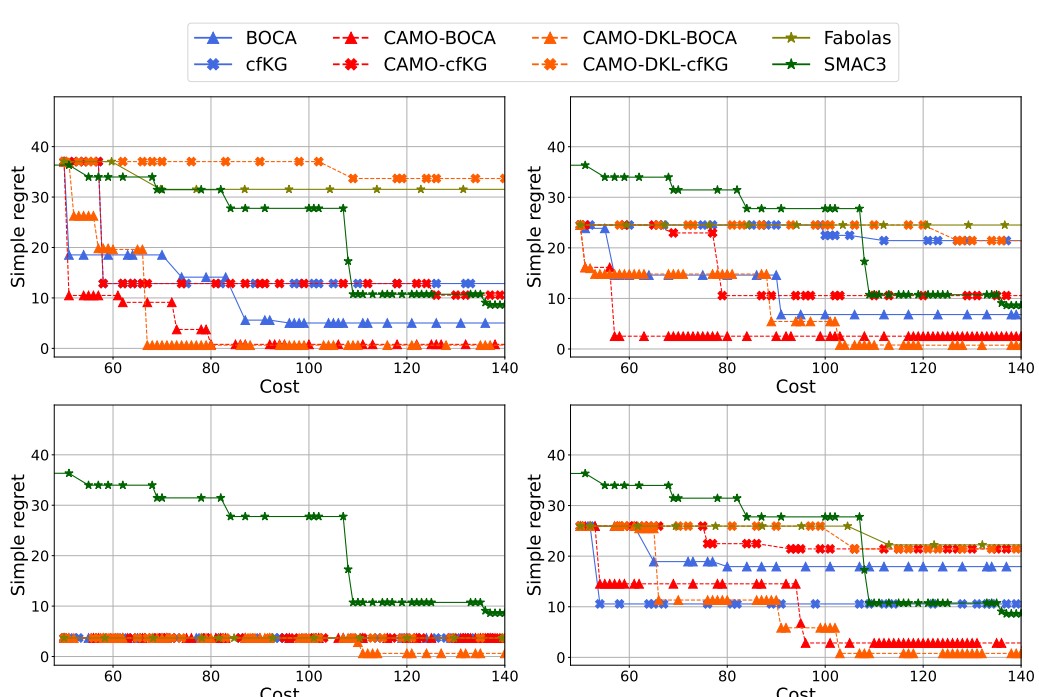

Figure 17: MFBO results from four seeds on Bohachevsky function.

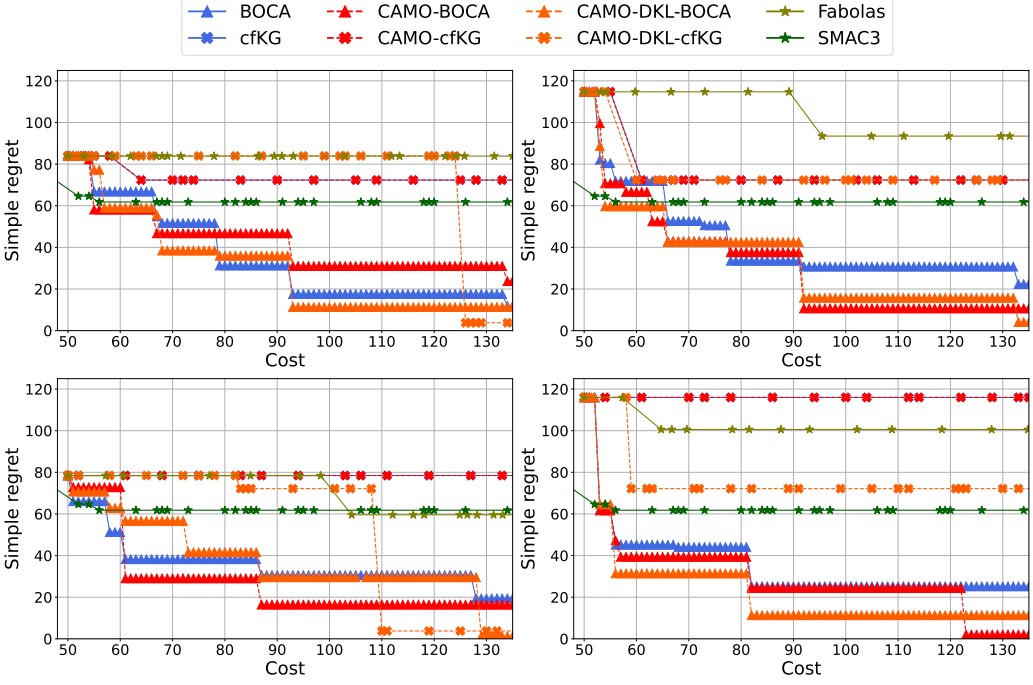

Figure 18: MFBO results from four seeds on Borehole function.

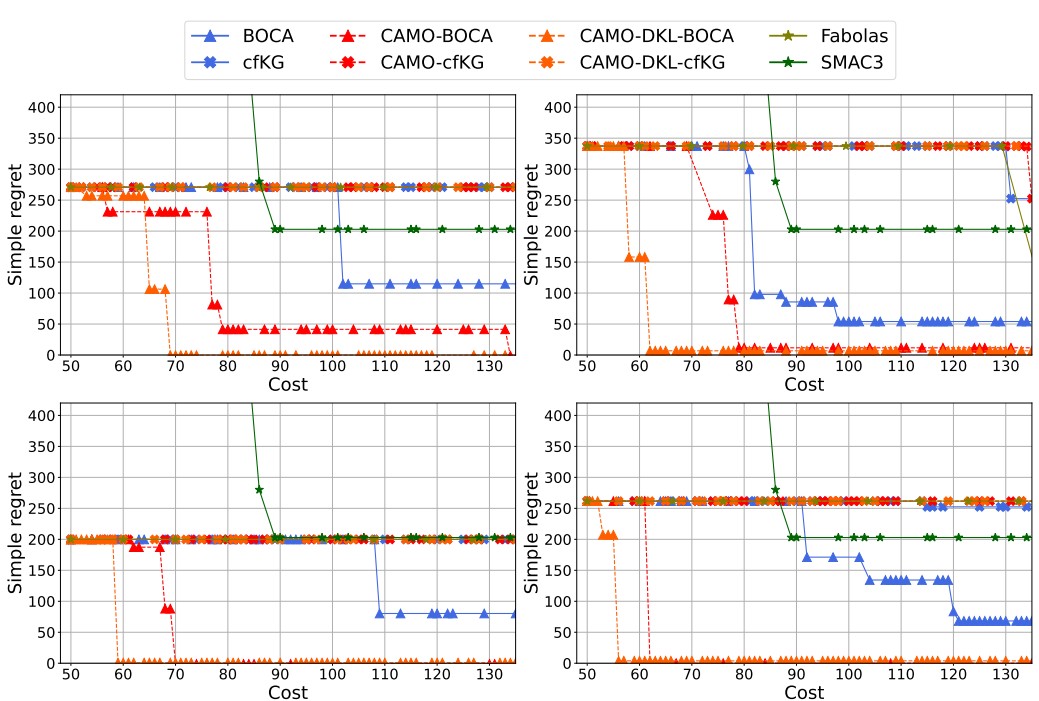

Figure 19: MFBO results from four seeds on Colvile function.

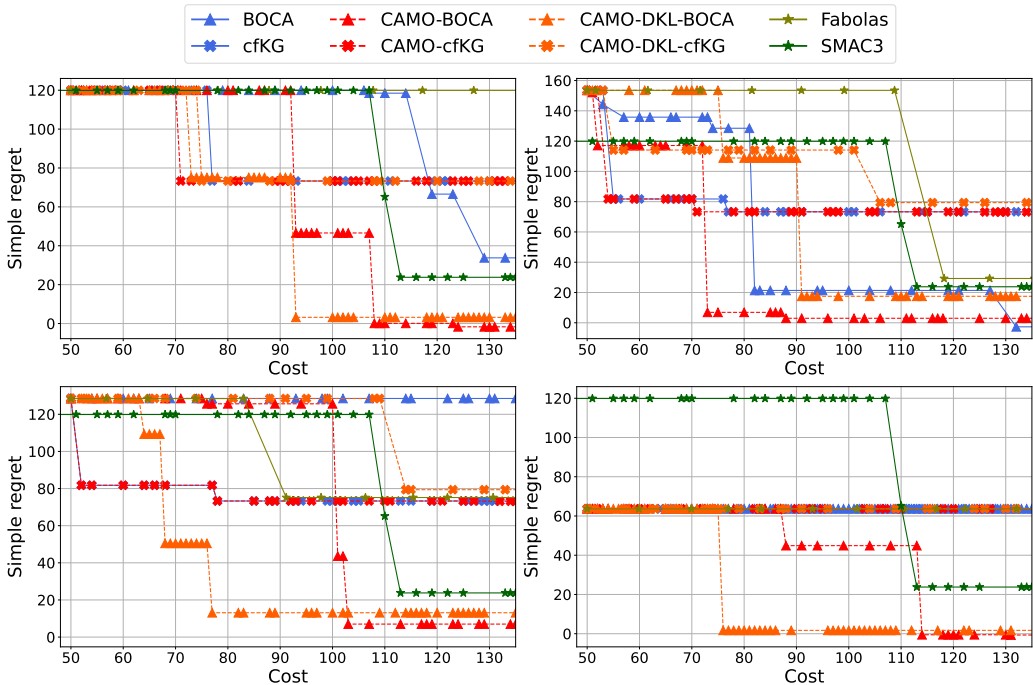

Figure 20: MFBO results from four seeds on Himmelblau function.

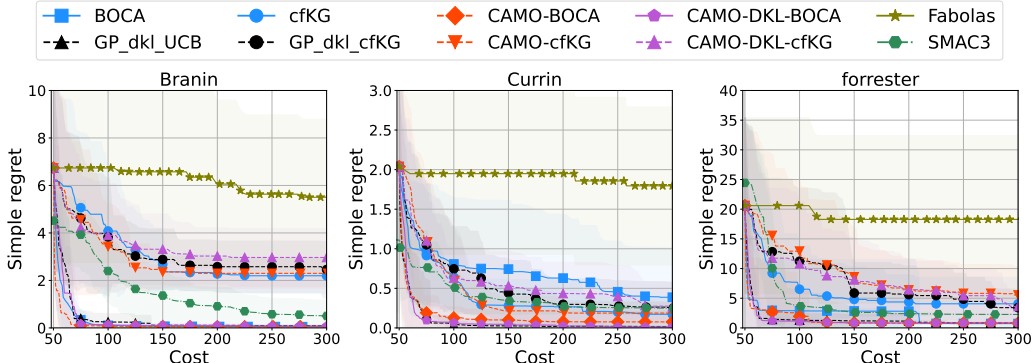

Figure 21: BOCA-DKL explore on Branin Currin and forrester dataset.

## SUPPLEMENTARY REFERENCES

Wei Xing, Yuxin Wang, and Zheng Xing. Continuar: Continuous autoregression for infinite-fidelity fusion. Advances in Neural Information Processing Systems, 36, 2024.

Loic Le Gratiet and Claire Cannamela. Cokriging-based sequential design strategies using fast cross-validation techniques for multi-fidelity computer codes. Technometrics, 57(3):418–427, 2015. doi: 10.1080/00401706. 2014.928233.

