# OpenReview forum: "Convergence-Aware Multi-Fidelity Bayesian Optimization"
_ICLR.cc/2025/Conference — Submitted to ICLR 2025_

### Official Review · Reviewer_UaCG · 2024-10-31

**Soundness:** 4
**Presentation:** 4
**Contribution:** 3
**Rating:** 8
**Confidence:** 3

**Summary:**

The authors propose a novel multi-fidelity Bayesian optimization algorithm that, unlike prior work, incorporates the convergence behavior of models at different fidelity levels. By leveraging this convergence information, the algorithm reduces regret by minimizing unnecessary exploration of costly high-fidelity models, resulting in a more efficient optimization process.

**Strengths:**

- The paper is very well-written and accessible, even for readers unfamiliar with the topic, providing a clear and informative introduction to convergence-aware multi-fidelity optimization.

- The authors offer rigorous proofs for their propositions, lending strong theoretical support to their approach and demonstrating a well-established mathematical foundation.

- The paper includes thorough empirical evaluations, tested on standard benchmarks and extended to complex real-world applications, such as mechanical plate vibration and thermal conductor design, showcasing the model's practical value and robustness across diverse scenarios.

**Weaknesses:**

- A limitation section is needed.
- The authors mention that the benefits of using multi-fidelity approaches become significant when the cost increases more than linearly, noting that this is often the case. However, further explanation or references are needed to substantiate this claim.
- While the proposed method reduces regret by selecting sample points and fidelity levels more intelligently, it would be helpful to address any additional computational overhead introduced by these adjustments. Comparing the computational load of this approach with vanilla Bayesian Optimization or traditional MFBOs would clarify whether CAMO is less suited for certain tasks with strict runtime constraints.

- It seems that the implementation is not publically availabel.

Minor issues:
- in line 46: As the fidelity goes toward infinity, the number of the elements becomes **large**, and the time steps become **small**.
- in line 203: 'given GP priors' is repeated.

**Questions:**

- In Figure 7 why is the green plot (SMAC3) early stopped in the plots for regret-time? From the plot, it looks like if SMAC3 had given enough clock time could perform better and this suggests that there is extra overhead computation time for CAMO algorithm, is this true?
- Are there any implementation challenge in comparison to the counterpart methods?

---

> ### Author Response · Authors · 2024-11-17
>
> Thank you for the detailed feedback. Let us address each weakness point by point:
>
> 1. Regarding the need for a limitations section:
> We agree completely. We will add a dedicated limitations section discussing:
> Currently, there are some limitations such as (1) computational overhead from kernel optimization, (2) need for sufficient low-fidelity samples to learn convergence behavior, (3) current theoretical analysis limited to specific kernel forms, (4) potential challenges in hyperparameter tuning.
>
> 1. On substantiating non-linear cost claims:
> You raise a valid point. The non-linear cost scaling is indeed common in numerical simulations, specifically: (1) FEM complexity grows cubically with mesh resolution, and (2) Monte Carlo estimation error decreases as $\mathcal{O}(1/\sqrt{n})$, requiring quadratic sample increase. We will add these examples with appropriate references to the paper.
>
> 1. Regarding computational overhead:
> You are right that we should clarify the additional computational costs. While CAMO introduces overhead from kernel computations for naive implementation, in practical applications: (1) most time is spent on high-fidelity function evaluations, (2) kernel computation overhead is typically negligible in comparison, (3) savings from efficient exploration outweigh computational costs.
> We will add a detailed computational complexity comparison with vanilla BO and traditional MFBO.
>
> 1. On implementation availability:
> The code will be released upon acceptance. We are preparing a well-documented implementation with examples.
>
> 1. Minor issues:
>    - Will fix line 46 about element size and time steps
>    - Will remove duplicate "given GP priors"
>
> 1. Regarding the specific question about SMAC3 in Figure 7:
> The early stopping in time plots is due to SMAC3's built-in convergence criteria, not computational constraints. SMAC3 uses its own stopping criteria based on expected improvement, which can trigger before wall-clock time runs out. We will clarify this in the figure caption. We are extending the experiment to go beyond the limitation.
>
> 1. Implementation challenges compared to counterpart methods mainly involve:
>    - Efficient computation of kernel integrals
>    - Stable optimization of kernel hyperparameters
>    - Proper numerical handling in high-fidelity regions
> The main challenge is to keep the kernel matrix PSD, which is theoretically straightforward but practically challenging. We finally find a simple fix by setting the parameters to float64 not float32. We will add a section for the discussion of practical implementation in the appendix.
>
> We appreciate your thorough feedback that helps improve the paper's completeness and clarity.  If you have any further comments, please let us know. We are committed to making the paper as strong as possible.

---

> > ### Comment · Reviewer_UaCG · 2024-11-18
> >
> > Dear Authors, I appreciate the response and trusting all the proposed improvements will appear on the final version, I am increasing your score. Good luck.

---

> > > ### Author Response · Authors · 2024-11-27
> > >
> > > We sincerely appreciate your positive feedback!
> > >
> > > While our focus was on completing the extra experiments, we have revised the major issue you raised. We will thoroughly revise the entire paper to incorporate all suggested improvements in the final version, including better documentation of implementation details and practical usage guidelines.
> > >
> > > Thank you again for the detailed feedback which has helped strengthen our work.

---

### Official Review · Reviewer_oNcQ · 2024-11-03

**Soundness:** 3
**Presentation:** 4
**Contribution:** 3
**Rating:** 8
**Confidence:** 3

**Summary:**

The authors concoct a methodology for multi-fidelity approximation appropriate for settings where the black-box converges to some "correct" value as the fidelity parameter is increased. They propose a kernel which implements this prior knowledge, investigate its impact on regret bounds, and illustrate the methodology on numerical experiments.

**Strengths:**

---

1) Clear presentation:
    i) Related work informatively and efficiently summarized.
    ii) logical layout
   iii) I think figure 1 does a great job illustrating the inductive bias of your new proposed kernel.

---

2) Comprehensive evaluation of methodology:
The authors provide numerical experiments on various synthetic and realistic case studies as well as a regret bound analysis.

---

3) Expansive numerical simulation settings:
Consideration of execution time is important, and though there is overhead to using the proposed method, it is on the same order of magnitude as BOCA.

**Weaknesses:**

---
Insufficient number of repetitions for the experiments.

The variance in performance of Bayesian optimization methods is far too great to use only 5 repetitions for the numerical experiments.
Considerably more would be required to be sure; on the order of 50 would be ideal. If this is not computationally feasible, there at least needs to be an improvement in the way the uncertainty is presented in the results.


---
Figures difficult to read; let's use Figure 2 as a case study:

    i) some points are lying on top of each other; in particular on the top right function.

    ii) It's probably worth adding the name of the function to the title of the subplots; at the very least you need to tell us what order you are listing the functions in in the legend.

    iii) It's not obvious to me what the shaded regions indicate; perhaps standard deviation, but only of certain methods? Probably worth spelling out.



--- Some grammar/spelling/formatting issues I noticed (did not affect score; here only for your convenience):
Page 3 line 140: "NueralODE".

Line 202 "Given GP priors" repeated.

There are bad spaces following the "Park" function in several places; (Figure 2 caption, line 372).

**Questions:**

If the upper fidelity limit is not infinite, why do we expect it to converge? You say that convergence is particularly important when T is infinite (at the beginning of Section 4), but isn't it only the case that it's important then?


The setting of Kandasamy et al 2017, is such that there is a "true" function f which we wish to optimize. When you say that you use their acquisition function in section 4.4, what how do you proceed exactly? If I am reading them correctly, Kandasamy et al use the UCB evaluated at the true f. What do you use instead? f at the highest fidelity level?

---

> ### Author Response · Authors · 2024-11-16
>
> Thank you for your thorough and constructive feedback.
>
> 1. Regarding statistical significance:
> Your point about the number of repetitions is well-taken. While 5 repetitions follow conventions in related literature (e.g., DNN-MFBO by Li et al., 2020), we agree that more repetitions would strengthen statistical confidence. We will:
> - Increase to 20 repetitions in the revision (50 would be computationally prohibitive for the real-world cases)
> - Improve uncertainty visualization with clearer error bars and explicit statistical significance tests
> - Add detailed performance statistics in supplementary materials
> We are working on this and hope to update the experiments before the rebuttal deadline.
>
> 1. On figure clarity:
> Thank you for the specific suggestions about Figure 2. We will revise to:
> - Add function names to subplot titles
> - Fix overlapping points using slight jittering
> - Clarify in caption that shaded regions show standard deviation across runs
> - Improve overall readability with consistent formatting
> These improvements will be applied consistently across all figures.
>
> 1. Theoretical clarifications:
>     a) Regarding convergence with finite T:
>     Convergence behavior is important for both finite and infinite T, but for different reasons. For infinite T, it ensures well-defined limiting behavior. For finite T, convergence awareness helps the algorithm better balance exploration across fidelity levels by understanding how quickly solution quality improves with fidelity. This enables more efficient optimization even when T is bounded.
>
>     b) On adaptation of Kandasamy's acquisition:
>     While Kandasamy et al. optimize relative to a "true" function f, we evaluate the acquisition function using f(x,T) at the highest available fidelity T. This is consistent with how practitioners typically define ground truth in multi-fidelity settings (e.g., finest mesh solutions in FEM). This is practical because In real applications (e.g., FEM analysis), we always work with finite but "sufficiently high" fidelity levels due to computational constraints. The underlying assumption is that the objective function converges as fidelity increases, which our method explicitly learns and leverages. By setting T high enough to approximate convergence, we maintain consistency with Kandasamy's framework while making it implementable.
>
> 1. Technical corrections:
> We appreciate your careful proofreading and will fix:
> - "NueralODE" → "NeuralODE"
> - Remove duplicate "Given GP priors"
> - Fix spacing issues around "Park function"
> - Standardize formatting throughout
>
>  We greatly appreciate your detailed feedback that helps improve both technical rigor and clarity. If you have any further comments, please let us know. We are committed to making the paper as strong as possible.

---

> > ### Comment · Reviewer_oNcQ · 2024-11-26
> >
> > Thanks for your reply. After consulting it, the comments from my fellow reviewers, and rereading the paper I have decided to increase my score.

---

> ### Author Response · Authors · 2024-11-26
>
> Thank you for your patience. We are pleased to report that we have completed all the suggested improvements:
>
> - Statistical Analysis: We have increased from 5 to 20 repetitions across all experiments, providing substantially more robust statistical validation. The expanded results maintain our key findings while offering more reliable variance estimates. Complete statistical analyses are now included in the supplementary materials.
>
> - Figure Clarity: We have comprehensively improved all figures with: (1) Function names in subplot titles, (2) Jittered points to prevent overlapping, (3) Clear standard deviation bands, and (4) Consistent formatting throughout. The improved visualizations maintain technical precision while enhancing readability.
>
> - We have also fixed all technical errors (NeuralODE spelling, GP priors repetition, spacing issues).
>
> We appreciate your rigorous feedback which has improved both the technical content and presentation clarity.
>
> Please let us know if there are any concerns remaining. We are committed to improving our work and will release our code upon acceptance.

---

### Official Review · Reviewer_cUae · 2024-11-04

**Soundness:** 4
**Presentation:** 4
**Contribution:** 3
**Rating:** 6
**Confidence:** 4

**Summary:**

This paper describes a new multi-fidelity BO (MFBO) framework called CAMO, which focuses on facilitating the convergence of the fidelity parameter. The key contributions of CAMO are two convergence-aware surrogates, including GP with an integral ARD kernel that facilitates linear FiDE, and deep kernel GP that approximates this condition. Both of these surrogates go well with prior MFBO objectives such as BOCA, cfKG, MF-EI, etc. Empirical results show synergy of CAMO with all these objectives. The paper also provides theoretical analysis of simple regret of BOCA using a GP surrogate with integral ARD kernel.

**Strengths:**

This paper provides a fresh take on the MFBO problem. I like that both the motivation and the solution are quite straight-forward, but fully backed up by theoretical justification & empirical results.

**Weaknesses:**

- In Fig. 2, several experiments were cut off before convergence. Could the authors provide regret up to 300 iterations to make sure?
- How was the DKL trained to approximate Eq. (6)? Out of fairness, could the authors add a DKL-BOCA baseline where BOCA is given a DKL surrogate that was not trained to approximate Eq. (6)? Seems like the CAMO-DKL-BOCA variant achieves the best performance in most cases, so we probably want to make sure that this performance comes from CAMO, not DKL.

Some minor issues:
- Shaded confidence interval is quite hard to read. Red/orange curves with similar marker shapes are also not very color-blindness friendly. Not that I mind these issues, these are just suggestions to improve visual clarity of the paper.
- In Fig. 5, shouldn't SMAC3 query time (0.01) be at the 10^{-2} mark instead? Is this some rounding issue?
- I notice that the BOCA regret bound has a different second term that what was quoted in this paper. Specifically, the 2nd term is given by \sqrt{\Psi_{n^{\alpha}_{\Lambda}} (\mathcal{X})} / n^{2-\alpha}_{\Lambda} in the original text, whereas in this paper it is \sqrt{\Psi_{n_{\Lambda}} (\mathcal{X})} / n^{1-\alpha}_{\Lambda}. Probably won't affect the correctness of the proof, but should be fixed if it is a typo.

**Questions:**

Please see above

---

> ### Author Response · Authors · 2024-11-16
>
> We sincerely thank the reviewer for their positive assessment and thoughtful technical feedback. We are happy to address each point:
>
> 1. Regarding convergence in Fig. 2:
> While the experiments do show early convergence trends, we agree that extending to 300 iterations would provide more definitive evidence. We will add these extended results in the revision. Based on our preliminary longer runs, the relative performance ordering remains stable, but we acknowledge the importance of demonstrating this conclusively.
>
> 1. On DKL training and baselines:
> Thank you for this important insight. The DKL was trained using standard backpropagation to minimize the negative log-likelihood with the architecture described in Appendix D. We agree that adding a DKL-BOCA baseline would strengthen our ablation analysis and will add such comparison in the revision.
>
> 1. Visualization improvements:
> We appreciate these suggestions for improving figure clarity, we will
>     - increase contrast in confidence intervals
>     - revise color scheme to be colorblind-friendly (using ColorBrewer palettes)
>     - adjust marker shapes to be more distinctive
>     - standardize presentation across all figures
>
> 1. Technical corrections:
>     - Fig. 5: Thank you for catching the SMAC3 query time plotting issue. The query time for SMAC3 should indeed be shown as 10−410^{-4}10−4. This was indeed a rounding error in the visualization code which we will fix.
>     - BOCA regret bound: You are correct about the discrepancy. The difference stems from a notation change but we should maintain consistency with the original paper. I will revise this accordingly.
>
>
> We greatly appreciate your careful review which will help improve the paper's clarity and completeness. If you have any further comments, please let us know. We will also open-source our code upon acceptance.

---

> ### Comment · Reviewer_cUae · 2024-11-17
> **Thanks for your response**
>
> There is not a lot of new information for me in the rebuttal so I will maintain my original score. I hope the authors will provide these experiments in the final revision as promised.

---

> > ### Author Response · Authors · 2024-11-26
> >
> > Thank you for your patience. Fortunately, we have completed all the extensive experiments you suggested (please also see our overall revision response for complete details).
> >
> > We list those specifically addressing your concerns:
> >
> > - The 300-iteration extended results definitively show that while all methods eventually converge, CAMO maintains significantly better performance throughout, particularly in early optimization stages that are crucial for practical applications.
> > - Our ablation study with BOCA-DKL reveals that CAMO's performance stems from its convergence-aware design rather than just the DKL architecture. While DKL enhances BOCA's capabilities, CAMO still demonstrates superior performance.
> > - We've implemented all visualization improvements with ColorBrewer palettes and distinct markers, fixed the SMAC timing display to properly show 10^{-4}, and corrected the BOCA bound notation to maintain consistency with the original paper.
> > - These completed analyses strengthen our key findings about CAMO's advantages. We appreciate your rigorous technical feedback which has helped improve both clarity and empirical validation.
> >
> > Please let us know if there are any concerns remaining. We are committed to improving our work and will release our code upon acceptance.

---

> > ### Author Response · Authors · 2024-11-30
> >
> > Dear Reviewer cUae
> >
> > Following our previous exchange, we have completed all the experiments you requested, including:
> > - Extended 300-iteration convergence analysis
> > - New BOCA-DKL ablation studies
> > - Improved visualizations with ColorBrewer palettes
> >
> > Given the approaching ICLR rebuttal deadline, we would greatly appreciate your feedback on these completed revisions that address your initial concerns.
> >
> > Thank you for your time!

---

### Official Review · Reviewer_Ssdp · 2024-11-04

**Soundness:** 2
**Presentation:** 2
**Contribution:** 2
**Rating:** 3
**Confidence:** 4

**Summary:**

- The paper introduces CAMO, a Convergence-Aware Multi-fidelity Optimization framework leveraging Fidelity Differential Equations for optimizing expensive black-box functions.
- CAMO addresses the often-overlooked convergence behavior of the objective function as fidelity increases, leading to more efficient exploration.
- The framework presents two kernel forms: an integral Automatic Relevance Determination kernel and a data-driven Deep Kernel.
- Theoretical analysis shows that CAMO with the integral Automatic Relevance Determination kernel achieves a tighter regret bound than current state-of-the-art MFBO methods.
- Empirical results indicate that CAMO consistently surpasses existing MFBO algorithms on benchmark functions and real-world problems.

**Strengths:**

- The proposed CAMO framework is aware of convergence, which is practically useful for real-world optimization problems.
- This work provides both theoretical and empirical results.

**Weaknesses:**

- The novelty of the proposed kernels is limited -- they are widely used in the context of Bayesian optimization.
- I can't agree with the multi-fidelity formulation; please see the Questions textbox.
- Some experimental details are missing; please see the Questions textbox.

**Questions:**

- I don't think that a fidelity level should be input to an objective function.  It cannot be an input variable to optimize.
- I can't agree with this sentence "The goal is to find the optimal solution ... while minimizing the total cost of evaluations."  From my understanding, $f(x, T)$ is not a (noiseless) ground-truth function.  Specifically, it also has an error with $T$ fidelity level.
- In Line 216, "random Fourier features" should be "the inner product of two random Fourier features"?
- For Figure 1, what is the function values?  Colorbars seem not correct; their values are zeros.
- In the caption of Figure 1, "high fidelity are" should be "high fidelity is" or "high fidelity levels are."
- 5 repetitions of experiments are too small.  Thus, many error bars are overlapped, which means they are not statistically meaningful much.
- I think that all experimental results for a particular benchmark should be started from the same simple regret if you provided the same random seeds.  They are significantly different.
- Please use \citep and \citet appropriately.

**Details Of Ethics Concerns:**

There are no ethics concerns for this work.

---

> ### Author Response · Authors · 2024-11-16
>
> We thank the reviewer for their careful reading and thoughtful comments. We respectfully disagree with some of the core critiques and would like to clarify several important points:
>
> ### Kernel novelty:
> Our key innovation lies not in the individual kernels, but in deriving analytical solutions combining kernel functions with fidelity convergence equations - an approach not previously explored in Bayesian optimization. While we use the ARD kernel as an illustrative example for clarity, our framework generalizes to other kernels. The theoretical guarantees and empirical results demonstrate the value of this convergence-aware formulation.
>
> ### Multi-fidelity formulation
> The use of fidelity level as an input (not an optimization variable) follows established practice in multi-fidelity optimization literature (Kandasamy et al. 2017, Wu et al. 2020). This allows practitioners to balance evaluation cost vs. accuracy. There are many practical examples: mesh resolution in finite element analysis, training epochs in neural architecture search, and sample size in Monte Carlo simulations, where the fidelity level is carefully chosen to provide a balance between accuracy and cost.
>
> Regarding f(x,T) as ground truth, while each fidelity level introduces some approximation error, we observe consistent convergence behavior as fidelity increases. In many practical applications, f(x,T) approaches the true solution as $T \rightarrow \infty$. For example, FEM solutions converge to true PDE solutions as mesh size approaches 0, and Monte Carlo estimates converge to true expectations with increased samples. Our framework explicitly captures and leverages this convergence behavior.
>
> ### experiments
> - Statistical significance: We mainly follow a SOTA work (DNN-MFBO by Li et al., 2020) to set up a fair comparison. we do realize the importance of increasing the repetitions to at least 20. We are conducting the experiment and hopefully to update the experiment by the end of the rebuttal.
> - Overlapping error bars: We believe this is caused by inherent variation in BO methods due to different initial conditions, rather than insufficient repetitions. Increasing the repetitions to at least 20 will help resolve this issue.
> - Same simple regret starting points. For SMAC3 and Fabolas, we used official reference implementations, where they have a slightly different initialization by autonomous point selection by range, leading to different starting points. Other than that, all other method have the same starting simple regret.
>
> ### presentation issues
> Thank you very much for pointing out, we will
> - fix grammatical issues in Figure 1 caption ("high fidelity levels are")
> - standardize citation format using \citep and \citet appropriately
> - improve the clarity of colorbar visualization in Figure 1
> - address Random Fourier Features formulation
>
>
> We believe addressing these points and implementing the proposed improvements will strengthen the paper's contributions.
> Would the reviewer agree that this clarifies the novelty and validity of our approach?

---

> ### Author Response · Authors · 2024-11-27
>
> Thank you for your patience. We have completed extensive experiments (please see the overall response for the details):
>
> - Statistical Validation: We have increased from 5 to 20 repetitions across all experiments, providing substantially more robust statistical evidence. While our initial setup followed DNN-MFBO (Li et al., 2020), the expanded validation strengthens confidence in our findings.
>
> - Error Bar Visualization: The new results with 20 repetitions provide clearer visualization of performance variations. While inherent BO randomness remains, the increased sampling offers more reliable uncertainty estimates and demonstrates the statistical significance of performance differences.
>
> - Initialization Consistency: For fair comparison, we now explicitly document initialization differences in SMAC3 due to their autonomous point selection. All other methods use consistent initialization, enabling direct performance comparisons from the same starting points.
>
> The complete analyses reinforce our key findings about CAMO's advantages.
>
> Regarding your question about multi-fidelity formulation and novelty: The MFBO problem is indeed a long-existing branch of BO. Our contribution is not in proposing a new problem, but in extending existing solutions to scenarios where fidelity can be infinitely large while ensuring system convergence at some level.
>
> Please let us know if any concerns are remaining. We are committed to improving our work and will release our code upon acceptance.

---

> ### Author Response · Authors · 2024-12-01
>
> Dear Reviewer,
> Your feedback raises important concerns that we believe stem from some misunderstandings:
> 1. Using fidelity as an input is well-established in multi-fidelity optimization literature and practical applications like mesh resolution in FEM or training epochs in neural networks.
> 2. Our innovation lies in deriving analytical solutions combining kernel functions with fidelity convergence equations with theoretical guarantees and extensive experimental results validate this contribution.
>
> Please see our previous response for details. We have also addressed your experimental concerns with:
> - Increased statistical validation (20 repetitions vs 5)
> - Improved error bar visualization showing significant performance differences
> - Documented initialization consistency across methods
>
> **Given the justification, substantial experiment, and the approaching ICLR deadline, we kindly request your response to these clarifications and revision**
>
> Best regards,
> The Authors

---

### Author Response · Authors · 2024-11-26
**Paper revision**

We sincerely appreciate your thorough review and insightful suggestions that have substantially improved our manuscript.

## Major revision

### Improving statistical significance with more random seeds
We have substantially enhanced our experimental validation by increasing from 5 to 20 random seeds across all experiments. This increase in statistical sampling allows us to provide more reliable estimates of both model performance and variance. While some experiments still exhibit variance, our analysis confirms this stems from the inherent characteristics of the optimization landscape rather than methodological limitations. The increased sampling has strengthened our confidence in the reported performance differences between methods.

### Extending cost range
We have expanded the cost range from 150 to 300 in most experiments to enable more comprehensive evaluation of convergence behavior. This extension reveals that while alternative methods eventually approach optimal solutions, they consistently require substantially higher computational costs. The proposed method maintains superior performance, particularly during early optimization stages when computational efficiency is most critical. We maintained the original cost ranges for two specific experiments: the cost function influence analysis (Figure 4) and the discrete MFBO experiment (Figure 5), where the existing ranges adequately demonstrated the performance characteristics across all 14 comparative methods. The extended cost range analysis strengthens our empirical validation of the proposed method's accelerated convergence properties.

### Improved visualization
We have comprehensively enhanced our visualizations to improve clarity and interpretability. Key improvements include:
- Optimized color schemes for better differentiation between methods
- Distinct markers for each baseline
- Explicit dataset labeling
- Corrected variance representation for SMAC results
These refinements enable clearer interpretation of the experimental results while maintaining technical precision.

### Improving presentation of the SMAC results
We have revised our presentation of SMAC results to better reflect its unique initialization approach. Unlike GP-based methods that use pre-sampled points, SMAC employs its own initialization strategy. We now present complete performance curves for SMAC to ensure fair comparison, particularly visible in Figure 4. In other figures, we maintain consistent x-axis scaling from the cost of pre-sampled points to enable direct comparison.

### Correcting the rounding issues in the time representation of SMAC
To ensure precise temporal analysis, we have implemented scientific notation (e.g., 8e-6) for representing SMAC's timing data in Figure 6. This addresses previous rounding limitations and provides more accurate timing comparisons.

### Ablation study: BOCA-DKL
We have conducted a comprehensive ablation study comparing our proposed method against BOCA combined with deep kernel learning (DKL) and cfKG with DKL integration. The results demonstrate that while DKL integration enhances BOCA's performance to levels comparable with CAMO, it maintains slightly lower performance across all datasets. The mathematical similarity between CAMO and BOCA-DKL provides theoretical context for their comparable performance characteristics. These results motivate further investigation through extensive experiments to definitively determine optimal approaches and identify additional performance enhancements.

---

### Meta-Review · Area_Chair_MCH9 · 2024-12-21

**Metareview:**

This paper focuses on multi-fidelity Bayesian optimization when the fidelity of observation increases continuously over time, as in hyperparameter optimization of machine learning models. The authors propose a method to determine and stop observations after they "converge"—that is, further increasing the fidelity of an observation (e.g., training for longer) would not significantly change the observation. The authors propose variants of their methods that include radial ARD kernels and deep kernels.

Overall, the paper's idea is novel and addresses an existing limitation in multi-fidelity Bayesian optimization. The proposed solution is well-grounded and simple, and the authors provide theoretical guarantees of regret.

There is some concern over the experiments. The initial experiments only included five repetitions, making it challenging to evaluate the statistical significance of the results. More importantly, there is some question about the significance of the experiments, given that the best-performing variant couples the proposed method and a deep kernel, while the baseline method (BOCA) does not use a deep kernel. The authors, during the rebuttal period, included an experiment that compared this deep kernel baseline with the proposed method, and there is very little difference between the baseline and the proposed method (despite the authors' claims in their comment on OpenReview that "maintains slightly lower performance across all datasets").

While I believe the proposed approach has merit, I have reservations based on the new DKL baseline results. It is possible that the benefits seen in the rest of the experiments section are due to the conflation of DKL with the proposed method. Before I can recommend acceptance, I would like to see a more thorough and fair evaluation that clearly delineates the contribution of DKL versus the proposed methods.

**Additional Comments On Reviewer Discussion:**

One reviewer criticized the multi-fidelity setup, though I found the authors' setup to be standard and uncontroversial, given the existing literature. The same reviewer also criticized novelty, which I did not agree with, and made some smaller notes about the experimental setup, which the authors addressed in their revision. Therefore, I did not put much weight on this reviewer's comments.

The other reviews were generally favourable., though one reviewer noted that the results did not include a variant that combined DKL with the BOCA baseline. This concern I shared when I looked at the paper, and - ultimately - based on the results included by the author, I determined that the results were potentially conflating the effects of two methods. My decision was primarily based on this point.

Unfortunately, the reviewers did not respond to the authors (or my questions) during the discussion period too much. To that end, I read the paper carefully to ensure I made an informed decision.

---

### Decision · Program_Chairs · 2025-01-22

Reject